# DUAL-STRUCTURE SELF-DISTILLED LEARNING FOR ENHANCING UNSUPERVISED SEMANTIC SEGMENTATION

## ABSTRACT

Unsupervised semantic segmentation (USS) aims to assign semantic labels to pixels without human annotations, yet existing methods struggle to capture semantic structures across different abstraction levels. We propose Dual-Structure Self-Distilled Learning (DS$^2$DL), a novel framework that performs dual self-distillation scheme that transfers semantic cues from the label space into the feature space, and aligns global semantic prototypes with local predictions to enhance both global coherence and local consistency. DS$^2$DL integrates two complementary structures:(1) an affinity structure that mines pairwise relationships through binary pair classification over similarity scores, while leveraging a reversed directional mining strategy to preserve fine-grained local consistency. (2) a cluster structure that derives semantic codes from global prototypes and aligns per-pixel predictions through Local-to-Global Code Prediction to enforce consistent global grouping. By jointly modeling both structures, DS$^2$DL enforces semantic consistency at both local and global levels, resulting in coherent and robust segmentation. Our method achieves substantial improvements over the strong baseline STEGO, with accuracy and mIoU gains of +16.7 and +3.3 on COCO-Stuff, +14.8 and +3.2 on Cityscapes, and +8.2 and +11.5 on Potsdam-3, respectively.

## 1 INTRODUCTION

Semantic segmentation aims to assign a semantic label to each pixel in an image, playing a critical role in scene understanding tasks such as autonomous driving Muhammad et al. (2022); Li et al. (2023a), robotics Nilsson et al. (2021); Rückin et al. (2024), and medical imaging Seibold et al. (2022); Karayegen & Aksahin (2021). While recent advances in deep learning have significantly improved segmentation performance, these models typically require large-scale pixel-level annotations, which are expensive and time-consuming to obtain. To overcome this limitation, Unsupervised Semantic Segmentation (USS) has emerged as a promising alternative, seeking to learn pixel-level semantics without relying on human-annotated labels.

In the absence of manual labels, unsupervised semantic segmentation (USS) relies on the assumption, known as semantic consistency, that pixels or patches within the same semantic region share similar visual patterns or contextual cues. Consequently, USS involves not only discovering latent clusters in the feature space, each corresponding to a semantic concept, but also accurately assigning each pixel to its appropriate semantic category.

Although USS has received relatively limited attention compared to its supervised counterpart, recent advances have proposed a variety of methods to achieve semantic segmentation without any annotated labels. These approaches leverage strategies such as visual consistency maximization Ji et al. (2019); Hwang et al. (2019), contrastive learning Seong et al. (2023); Ke et al. (2022), multi-view equivalence Hyun Cho et al. (2021), or some priors, including saliency prior Van Gansbeke et al. (2021), smoothness prior Lan et al. (2023). In particular, recent works Hamilton et al. (2022); Yin et al. (2022); Zadaianchuk et al. (2023); Li et al. (2023b); Sick et al. (2024); Kim et al. (2024); Lan et al. (2023) have made significant strides in advancing unsupervised semantic segmentation (USS), enabling superior performance in unsupervised semantic segmentation tasks. These methods harness the self-supervised inductive bias inherent in DINO-pretrained ViT models, which enables

the learning of pixel-wise semantically coherent object representations essential for unsupervised segmentation tasks.

Despite recent progress in unsupervised semantic segmentation (USS), most existing methods do not explicitly and jointly model the cluster and affinity structures of the feature space. The cluster-level structure promotes alignment between pixel representations and semantic prototypes, facilitating semantic grouping. In contrast, the affinity structure captures pairwise semantic similarities between pixels, encouraging spatial coherence and preserving object boundaries, thereby facilitating more accurate semantic assignment in subsequent stages. To address this limitation, we propose Dual-Structure Self-Distilled Learning ($DS^2DL$), a novel framework for unsupervised semantic segmentation that jointly captures both the affinity and cluster-level structures of the feature space. In contrast to traditional distillation paradigms that rely on external teacher models, Dual-Structure performs self-distillation within a single network by transferring hierarchical knowledge from label space to feature space.

Specifically, we design two complementary self-distilled structures:

- Affinity Structure Distillation: This module distills semantic information from label space into pixel-level feature space, ensuring local semantic consistency. We formulate the task as a binary pairwise similarity classification problem, determining whether two pixels belong to the same semantic region. Since ground-truth pairwise labels are unavailable, we approximate them using the cosine similarity between the corresponding learned label features. Notably, these label features tend to approximate one-hot vectors, inherently promoting high inter-class separability and intra-class compactness. This property enhances the reliability of similarity estimation and helps establish sharper boundaries between semantic regions.

- Cluster-Level Structure Distillation: This module adopts a Local-to-Global Code Prediction strategy to capture global semantic grouping. Specifically, semantic codes are derived from global prototypes and aligned with predictions obtained from local prototypes that serve as semantic anchors. This alignment enforces semantic consistency across spatial regions and training steps, providing a surrogate supervisory signal that encourages the emergence of globally consistent segmentation patterns.

## 2 RELATED WORKS

Self-supervised learning (SSL) has emerged as a powerful paradigm for learning visual representations without manual labels, using pretext tasks such as contrastive learning (e.g., MoCo He et al. (2020), SimCLR Chen et al. (2020)) or clustering-based methods (e.g., DeepCluster Caron et al. (2018), SwAV Caron et al. (2020)) to promote semantic discrimination and structure in the feature space. Recent generative approaches like MAE Wei et al. (2022) and SimMIM Xie et al. (2022) reconstruct masked image patches to learn rich contextual featuresWen et al. (2023), while methods like BYOL Grill et al. (2020) and DINO Caron et al. (2021) leverage momentum encoders for stable representation learning. These SSL-pretrained models have proven effective as backbones for unsupervised semantic segmentation (USS), where models segment images into semantic regions without annotations. Earlier clustering-based USS methods (e.g., IIC Ji & Vedaldi (2019), PiCIE Hyun Cho et al. (2021)) group feature embeddings into semantic clusters, while recent works like STEGO Hamilton et al. (2022), SmooSeg Lan et al. (2023), and EiCue Kim et al. (2024) leverage SSL features with additional constraints such as spatial smoothness, object-level contrast, or spectral cues for improved structure awareness. Complementary to these, knowledge distillation (KD) transfers knowledge from teacher to student models by aligning outputs or intermediate features Hinton et al. (2015b); Shen et al. (2019). Recently Self-distillation Zhang et al. (2019; 2022) is a variant of knowledge distillation that removes the need for an external teacher, instead leveraging a model's own predictions or features to provide internal supervision, thereby enhancing feature consistency and representation quality. Cross-modal distillation is proposed for unsupervised semantic segmentation that leverages synchronized LiDAR–image pairs for knowledge transfer Vobecký et al. (2025). Inspired by this, we propose Dual-Structure Self-Distilled Learning ($DS^2DL$), a novel self-distillation framework that explicitly models and distills both affinity and cluster structures from label space to feature space and from global to local, promoting semantic coherence and structure-aware learning for unsupervised segmentation.

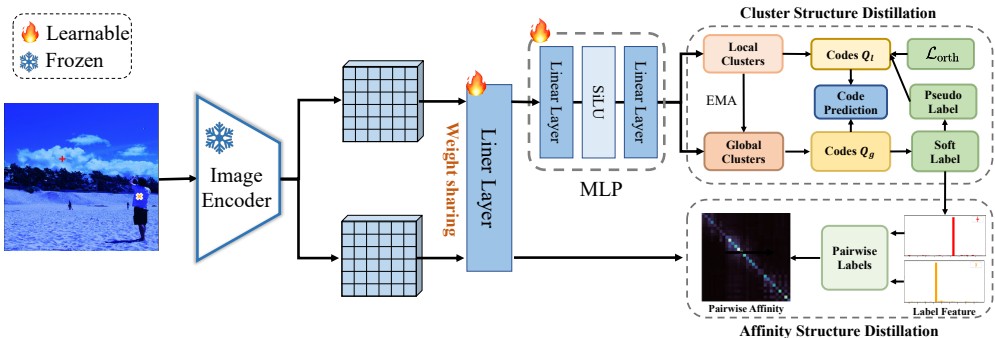

Figure 1: Overview of our DS$^2$DL framework. Our dual-structure framework consists of two separate branches: Cluster-Level Structure Distillation (CLSD) and Affinity Structure Distillation (ASD). ASD distills semantic relationships from learned label features into pixel-level representations to promote local semantic consistency, while CLSD adopts a Local-to-Global Code Prediction strategy to capture global semantic grouping.

## 3 METHODOLOGY

We first assume a binary relationship between pixels, where each pair either belongs to the same semantic cluster or to different ones. Based on this assumption, we reformulate the task of mining the affinity structure among pixels as a binary pairwise pixels classification problem. Since the true similarities between pixels are not known in advance, we adopt an adaptive strategy to identify informative pixel pairs for training by dynamically estimating their similarity.

To further capture global semantic grouping, we introduce a Local-to-Global Code Prediction strategy strategy that aligns semantic codes derived from global prototypes with predictions from local prototypes. This alignment promotes semantic consistency across spatial regions and serves as an implicit supervisory signal to guide the emergence of coherent segmentation patterns. An overview of the proposed method is provided in Figure 1. More details are given in the following subsections.

### 3.1 AFFINITY STRUCTURE LEARNING VIA BINARY PAIRWISE CLASSIFICATION

Given an arbitrary pair of pixels $x_i$ and $x_j$, we formulate pairwise similarity learning as a binary classification problem, which yields the following loss formulation:

$$\mathcal{L}_p = \mathbb{E}_{\{x_i,x_j\}\sim\mathcal{D}}\big\{y_{(x_i,x_j)} \cdot \log\left(1 + \exp\left(-\mathcal{S}(x_i,x_j) - b\right)\right)$$
$$+ (1 - y_{(x_i,x_j)}) \cdot \log\left(1 + \exp\left(\mathcal{S}(x_i,x_j) + b\right)\right)\big\} \quad (1)$$

Here, $y_{(x_i,x_j)} = 1$ if $x_i$ and $x_j$ belong to the same class, and $y_{(x_i,x_j)} = 0$ otherwise. $\mathcal{S}(x_i,x_j)$ is the similarity between $x_i$ and $x_j$, which is implemented as the normalized dot product, and $b$ is a learnable bias. The behavior of this formulation can be better understood through its decision boundary: $S(x_i,x_j) + b = 0$. When the similarity score $S(x_i,x_j)$ exceeds $-b$, the pair is predicted to belong to the same class; otherwise, it is considered a negative pair.

In general, two issues in Eq. (1) need to be addressed, namely, 1) the pairwise label $y_{(x_i,x_j)}$ is unknown during the USS process. The pairwise labels are commonly obtained by mining k-nearest neighbors in the feature space based on similarity $S(x_i,x_j)$. Nevertheless, this similarity-based k-NN mining often results in noisy labels, since similarity alone cannot perfectly capture class membership. 2) Constructing sample pairs is arguably one of the most critical factors influencing the performance of pairwise similarity learning. These two issues are addressed separately in the next Sections, respectively.

### 3.2 DISTILLING SEMANTIC AFFINITIES FROM LABEL SPACE

Given the unlabeled image dataset $\mathcal{I} = \{\mathbf{I}_i\}_{i=1}^n$ and the predefined number of semantic clusters $C$. For each pixel $x \in I_i$, the network outputs a probability distribution over $C$ semantic classes,

denoted as the label feature $\mathcal{L}_x = \{l_{cx} = q(c|x)\}_{c=1}^C$, where $q(c|x) \in [0, 1]$ represents the predicted probability that pixel $x$ belongs to class $c$. The label feature offers several important benefits for unsupervised semantic segmentation. First, it encodes semantic affinity in a soft and probabilistic manner, allowing for more nuanced representations than hard pseudo-labels Hinton et al. (2015a). Using soft label allows for better modeling of ambiguous or boundary pixels, which are often misclassified with hard pseudo-labels. Second, the label feature tends to form sparse or even one-hot-like distributions during training, which naturally facilitates clustering and semantic grouping across pixels. Third, these features are directly interpretable, as each dimension corresponds to a semantic class, making it easier to visualize, analyze, and supervise the learned structures. Therefore, We introduce a pairwise soft label, which captures the similarity between two pixels in the label feature space, defined as:

$$y_{(x_i, x_j)} = \mathcal{L}_{x_i} \cdot \mathcal{L}_{x_j} = \sum_{c=1}^C q(c \mid x_i) \, q(c \mid x_j). \tag{2}$$

Intuitively, $y_{(x_i, x_j)}$ serves as a soft guidance signal that assigns higher weights to likely true positive pairs while down-weighting potentially false positive ones. This pairwise soft label relies solely on the predictions from the global clusters representation, which is maintained via an Exponential Moving Average (EMA) update of the local clusters predictions.

## 3.3 SAMPLE PAIR CONSTRUCTION

Constructing effective sample pairs plays a pivotal role in the performance of pairwise similarity learning . We focus on two essential aspects of pair sampling: pair coverage and pair hardness.

**Positive/negative sample balance.** Because it is impossible to enumerate all the pixel-wise pair combinations in a large-scale dataset, when these pairs are constructed solely within individual images, they tend to produce an overwhelming number of positive sample pairs, as spatially adjacent pixels are more likely to belong to the same semantic class. This leads to a biased pair distribution, dominated by likely-positive pairs, which can hinder the model's ability to learn discriminative features across semantic boundaries. To address this issue, we introduce cross-image pixel pairing within each mini-batch images. By applying a near-derangement permutation Brualdi & Dahl (2025) at the image level, we ensure that each image is matched with a different one. This cross-image matching also enriches the diversity of the sampled pixel pairs. The resulting cross-image pixel pairs are more likely to span different semantic classes, which increases the proportion of negative pairs and rebalances the training distribution.

**Easy/hard sample mining.** Hard sample mining is essential for learning discriminative features in challenging vision tasks such as classification, detection, and segmentation. Difficult examples, especially near semantic boundaries, often drive effective representation learning, while ignoring them can lead to overfitting and poor generalization Shrivastava et al. (2016); Harwood et al. (2017); Ge et al. (2018); Chang et al. (2017); Wang et al. (2021).

In pairwise similarity learning, the decision boundary between positive and negative pairs is implicitly defined by $S(x_i, x_j) + b = 0$, where $b$ is a learnable bias. The signed distance $t = S(\cdot) + b$ quantifies confidence: smaller $t$ values indicate uncertain (i.e., harder) samples, while larger values denote confident predictions. We observe that hard positive pairs lie near the boundary with low similarity scores, whereas hard negative pairs are close to or across the boundary with high similarity. Thus, the directions of hard mining are inherently opposite for positive and negative pairs. Naively applying the same hardness weighting across both types may lead to conflicting gradients and ineffective optimization. To address this, we propose a Reversed Directional Mining strategy that modulates the gradient contributions in opposing directions: we assign lower weights to challenging positive pairs located close to class boundaries, while emphasizing more difficult negative pairs in these boundary regions. Specifically, we scale the positive pair loss using $1/r$, and the negative pair loss using $r$, where $r > 1$ is a single hyperparameter controlling the mining strength. This yields the following loss:

$$\mathcal{L}_p = \mathbb{E}_{\{x_i, x_j\} \sim \mathcal{D}} \big\{ y_{(x_i, x_j)} \log(1 + \exp(-t/r)) + (1 - y_{(x_i, x_j)}) \log\left(1 + \exp(rt)\right) \big\} \tag{3}$$

This formulation effectively emphasizes easy positives and hard negatives without gradient cancellation, resulting in more robust and discriminative representation learning. We define $Q_1(t) = \log(1 + \exp(-t/r))$ and $Q_2(t) = \log(1 + \exp(r \cdot t))$, where $r > 0$. Larger $r$ shifts $Q_1$ to

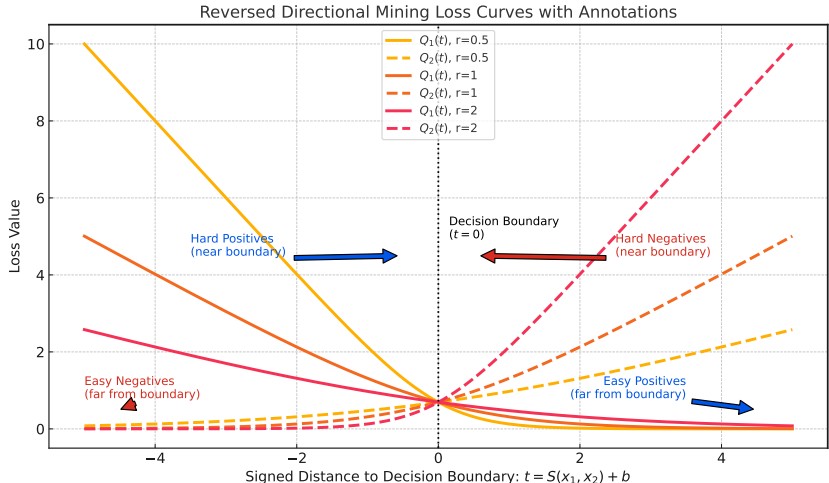

Figure 2: Visualization of Reversed Directional Mining. Loss curves for positive $Q_1(t)$ and negative pairs $Q_2(t)$ are shown, with sample hardness indicated by distance to the decision boundary.

**easy positive pairs** and $Q_2$ to **hard negative pairs**, and then plot their curves in Figure 2 to illustrate how they achieve hard pair mining of reverse directions.

### 3.4 CLUSTER STRUCTURE DISTILLATION VIA LOCAL-TO-GLOBAL CODE PREDICTION

Prior clustering-based methods for unsupervised segmentation operate offline, requiring repeated full-dataset passes and separate optimization stages Hyun Cho et al. (2021); Zadaianchuk et al. (2023). This decoupling often results in misaligned features and suboptimal segmentation. To address this, we adopt an online clustering strategy inspired by SwAV Caron et al. (2020), which integrates clustering objectives directly into training and supports continuous updates from streaming data.

More precisely, we introduce Local-to-Global Code Prediction, a mechanism that aligns local and global semantic structures through cross-prototype prediction. Specifically,for each pixel, we compute a pixel-wise semantic code by projecting global cluster prototypes onto the current image, and predict this code using local cluster prototypes derived from intra-image pixel groupings. Given a pixel feature $x$, we define its global semantic code as $q_g$, computed from global cluster prototypes $\{\tilde{c}_1, \ldots, \tilde{c}_C\}$, and its local prediction as $q_l$, obtained from local cluster prototypes $\{c_1, \ldots, c_C\}$. Our model is trained to predict $q_g$ using $q_l$, forming a Local-to-Global Code Prediction objective. The loss is defined as:

$$\mathcal{L}_{\text{L2G}}(x) = -\sum_{k=1}^{C} q_g^{(k)} \log q_l^{(k)}, q_l^{(k)} = \frac{\exp\left(\frac{1}{\tau} x^\top c_k\right)}{\sum_{k'} \exp\left(\frac{1}{\tau} x^\top c_{k'}\right)} \tag{4}$$

This local-to-global prediction framework can also be viewed as a form of self-distillation. Semantic knowledge is distilled by projecting pixel features onto a set of global cluster prototypes, which aggregate information across training samples. These global prototypes serve as a compact representation of dataset-level semantics, while the local cluster structures capture image-specific cues. By training the model to predict global codes from local ones, we effectively transfer the distilled global knowledge back to individual pixels, thereby promoting semantic consistency and improving generalization.

**Computing pixel-level codes online.** To enable online learning and capture semantic structure at the pixel level, we compute semantic codes independently within each image, using only its own pixel features. Specifically, given a single image with pixel features $\mathbf{X} = [x_1, \ldots, x_N] \in \mathbb{R}^{D \times N}$, we compute a semantic code $q_i \in \mathbb{R}^C$ for each pixel $x_i$ by matching it to a set of global cluster prototypes $\mathbf{C} = [\tilde{c}_1, \ldots, \tilde{c}_K] \in \mathbb{R}^{D \times C}$, which are shared across training and updated via exponential moving average (EMA).

To obtain the code matrix $\mathbf{Q} = [q_1, \ldots, q_N] \in \mathbb{R}^{C \times N}$, we solve the following optimization problem:

$$\max_{\mathbf{Q}} \mathrm{Tr}\left(\mathbf{Q}^\top \mathbf{C}^\top \mathbf{Z}\right) + \varepsilon \mathcal{H}(\mathbf{Q}) \quad \text{s. t.} \quad \mathcal{Q} = \left\{ Q \in \mathbb{R}_+^{C \times N} \middle| Q^\top \mathbf{1}_C = \mathbf{1}_N, Q\mathbf{1}_N = \tfrac{N}{C}\mathbf{1}_C \right\} \quad (5)$$

where $\mathcal{H}(\mathbf{Q})$ denotes the entropy of the assignment matrix, $\epsilon$ is a parameter that controls the smoothness of the mapping, and $\mathbf{1}_C$ denoting an $C$-dimensional vector of all ones. This entropy regularization encourages the assignments to be balanced across the $C$ prototypes. In doing so, it prevents a trivial solution where all pixels collapse to a single prototype, and instead promotes fine-grained and spatially diverse cluster assignments, enabling the discovery of small and localized semantic structures within the image.

We compute the soft assignment matrix $Q$ by solving an entropy-regularized optimal transport problem between pixel features and global prototypes. To solve this efficiently, we utilize Sinkhorn's algorithm Cuturi (2013), which iteratively normalizes the rows and columns of a similarity matrix to produce a doubly stochastic matrix with the desired marginal constraints. Empirically, three iterations are both efficient and adequate to obtain good results.

To enhance the distinctiveness of the cluster prototypes $\mathbf{C} = [\tilde{c}_1, \ldots, \tilde{c}_C] \in \mathbb{R}^{D \times C}$, we encourage their mutual orthogonality by enforcing $\mathbf{C}^\top \mathbf{C}$ to approximate the identity matrix. This constraint reduces redundancy among prototypes and ensures that each prototype captures unique semantic information. To this end, we introduce the following orthogonality regularization term:

$$\mathcal{L}_{\mathrm{orth}} = \left\| \mathbf{C}^\top \mathbf{C} - \mathbf{I} \right\|_1, \quad (6)$$

where $\| \cdot \|_1$ denotes the element-wise $\ell_1$-norm, and $\mathbf{I}$ is the identity matrix. This loss penalizes correlations between prototypes, promoting better separation and compactness in the semantic space.

## 3.5 LOSS FUNCTION

To enhance pixel-level discrimination and reinforce the model's semantic understanding, we employ a self-labeling strategy to provide supervision at the pixel level. During each training epoch, the model generates pseudo labels based on its current predictions. Let $\hat{y}_i \in \mathbb{R}^C$ be the predicted class probability vector for the $i$-th pixel, where $C$ is the number of semantic categories. Thus, we present the definition of the hard pseudo label and the corresponding pixel-wise cross-entropy loss in one equation:

$$y_i = \arg\max_k \hat{y}_{i,k}, \qquad \mathcal{L}_{\mathrm{data}} = -\sum_{i=1}^N \sum_{k=1}^C \mathbb{I}(y_i = k) \log \hat{y}_{i,k} \quad (7)$$

where $\mathbb{I}(\cdot)$ is the indicator function that outputs 1 if the condition is true and 0 otherwise. Minimizing $\mathcal{L}_{\mathrm{data}}$ encourages the model to reinforce its confident predictions, enabling effective self-supervised learning. We organize the loss into two coherent components: an *affinity-structure loss* $\mathcal{L}_p$ for pixel-wise relation learning, and a *prototype-based clustering loss* $\mathcal{L}_{\mathrm{clus}} = \mathcal{L}_{\mathrm{L2G}} + \mathcal{L}_{\mathrm{orth}} + \mathcal{L}_{\mathrm{data}}$ that captures global semantic structure. The overall objective is:

$$\mathcal{L}_{\mathrm{total}} = \mathcal{L}_p + \mathcal{L}_{\mathrm{clus}}. \quad (8)$$

## 4 EXPERIMENTS AND RESULTS

### 4.1 DATASETS

We conduct experiments on three publicly available datasets, following commonly adopted evaluation protocols Lan et al. (2023). **COCOStuff** Caesar et al. (2018) is a scene-centric dataset that contains 80 'thing' and 91 'stuff' categories. For evaluation, we adopt the standard 27-class setting, which merges the original labels into 15 thing and 12 stuff categories. **Cityscapes** Cordts et al. (2016) comprises street-view images collected from 50 cities, primarily focusing on urban driving scenes. Following standard practice, we use 27 semantic classes after removing the "void" category. **Potsdam-3** Ji et al. (2019) is a high-resolution remote sensing dataset containing 8,550 annotated images across 3 classes. We follow the default train/test split, with 4,545 images used for training and 855 for evaluation.

## 4.2 Evaluation Metrics

To compare clustering results with ground-truth annotations, we apply the Hungarian matching algorithm Kuhn (1955) to align predicted cluster assignments with semantic labels. Additionally, a fully connected Conditional Random Field (CRF) Krähenbühl & Koltun (2011) is applied as post-processing to refine the predicted segmentation maps, using the parameter settings from Lan et al. (2023). We report two commonly used metrics: mean Intersection over Union (mIoU) and pixel-level Accuracy (Acc), both averaged across all semantic classes.

## 4.3 Implementation Details

All models are implemented in PyTorch Paszke et al. (2019) and trained on two NVIDIA TiTAN XP GPUs. To ensure fair comparison with prior work Lan et al. (2023); Hamilton et al. (2022), we adopt a pre-trained DINO Caron et al. (2021) feature extractor with a ViT backbone trained on ImageNet. The feature encoder is frozen during training.

We adopt a lightweight architecture composed of a linear layer and a two-layer MLP with SiLU activations, together with two sets of prototype vectors initialized identically. We apply exponential moving average (EMA) updates to stabilize training, with the momentum coefficient set to $\alpha = 0.998$. The dimensionality of the output features is fixed at $D = 64$, and the temperature parameter used in contrastive terms is $\tau = 0.1$. To adapt to varying dataset complexities, we use Adam Kingma & Ba (2015) for the simpler Potsdam dataset, and RAdam Liu et al. (2020) for COCOStuff and Cityscapes, where Adam's unstable early updates are mitigated by RAdam's variance rectification, leading to more stable convergence on complex scenes. A batch size of 32 is adopted for all datasets. To enlarge the training set, a five-crop augmentation is employed for the Cityscapes and COCOStuff datasets. Training proceeds for 4,000 iterations on Potsdam-3, 8,000 iterations on Cityscapes, and 15000 iterations on COCOStuff. For reproducibility, we fixed the random seed to 123456 for COCO-Stuff, and to 0 for Cityscapes and Potsdam.

## 4.4 Comparison with State-of-the-Art Methods

To assess the effectiveness of our proposed method, we perform detailed comparisons with recent unsupervised semantic segmentation techniques, including both quantitative evaluation and qualitative inspection.

**Quantitative Evaluation.** The quantitative results on the three datasets are presented in Tables 1. On the COCO-Stuff dataset, using ViT-S/8 as the backbone, our proposed DS$^2$DL achieves remarkable improvements in unsupervised accuracy, outperforming recent methods such as STEGO Hamilton et al. (2022), HP Seong et al. (2023), SmooSeg Lan et al. (2023), EAGLE Kim et al. (2024), OMH Ozaydin et al. (2024), and D&S++ Vobecký et al. (2025). In particular, DS$^2$DL exceeds STEGO, HP, SmooSeg, EAGLE, OMH, and D&S++ by +16.7, +7.8, +1.8, +0.8, +7.4, and +8.2 in accuracy, and achieves mIoU gains of +3.3, +3.2, +1.1, +0.6, +2.5, and +1.3 respectively. On the Cityscapes dataset, with ViT-S/8 as the backbone, DS$^2$DL again demonstrates clear advantages over the same set of baselines. It surpasses STEGO, HP, SmooSeg, EAGLE, OMH, and D&S++ by +14.8, +4.5, +1.8, +2.8, +1.1, and +10.4 in accuracy, and by +3.2, +2.4, +2.4, +1.1, +2.6, and +1.0 in mIoU, respectively. On the Potsdam dataset, using ViT-B/8, DS$^2$DL achieves 85.2% accuracy and 74.1% mIoU, outperforming all prior methods. Compared to STEGO, HP, SmooSeg, EAGLE, OMH, and D&S++, it yields +8.2, +2.8, +2.5, +1.9, +2.5, and +4.9 higher accuracy, respectively. Its mIoU further improves upon STEGO (+11.5), SmooSeg (+3.8), EAGLE (+3.0), and D&S++(+5.0).

Our method achieves notable performance gains on COCO-Stuff (+16.7 Acc, +3.3 mIoU), Cityscapes (+14.8 Acc, +3.2 mIoU), and Potsdam-3 (+8.2 Acc, +11.5 mIoU) compared to the strong baseline STEGO. These improvements stem from the proposed dual-structure self-distilled framework, where the affinity structure models local pixel-wise relationships via binary classification, while the cluster structure imposes global semantic regularization through EMA-updated prototypes. The affinity structure models local pixel interactions via binary classification over pairwise similarity scores, enhanced with a reversed directional mining strategy for better spatial coherence. In parallel, the cluster structure, distilled through a Local-to-Global Code Prediction mechanism, provides global semantic guidance by aligning pixel features with prototypes. Together, these structures enable more accurate and structured segmentation.

Table 1: Performance on COCO-Stuff and Cityscapes (27 classes, ViT-S) and Potsdam-3 (ViT-B)

| Methods | backbone | COCOStuff | | Cityscapes | | Potsdam-3 | |
|---|---|---|---|---|---|---|---|
| | | Acc. | mIoU | Acc. | mIoU | Acc. | mIoU |
| IIC Ji & Vedaldi (2019) | R18+FPN | 21.8 | 6.7 | 47.9 | 6.4 | 65.1 | - |
| MDC/Deep Cluster Caron et al. (2018) | R18+FPN | 32.2 | 9.8 | 40.7 | 7.1 | 41.7 | - |
| PiCiE Hyun Cho et al. (2021) | R18+FPN | 48.1 | 13.8 | 65.5 | 12.3 | - | - |
| DINO Caron et al. (2021) | ViT/8 | 29.6 | 10.8 | 40.5 | 13.7 | 62.2 | 43.3 |
| + HP Seong et al. (2023) | ViT/8 | 57.2 | 24.6 | 80.1 | 18.4 | 82.4 | - |
| + TransFGU Yin et al. (2022) | ViT/8 | 52.7 | 17.5 | 77.9 | 16.8 | - | - |
| + STEGO Hamilton et al. (2022) | ViT/8 | 48.3 | 24.5 | 69.8 | 17.6 | 77.0 | 62.6 |
| + SmooSeg Lan et al. (2023) | ViT/8 | 63.2 | 26.7 | 82.8 | 18.4 | 82.7 | 70.3 |
| + OMH Ozaydin et al. (2024) | ViT/8 | 57.6 | 25.3 | 83.5 | 18.2 | 82.7 | - |
| + DepthG Sick et al. (2024) | ViT/8 | 55.6 | 26.7 | - | - | 80.4 | - |
| + EAGLE Kim et al. (2024) | ViT/8 | 64.2 | 27.2 | 81.8 | 19.7 | 83.3 | 71.1 |
| + D&S++ Vobecký et al. (2025) | ViT/8 | 56.8 | 26.5 | 74.2 | 19.8 | 80.3 | 69.1 |
| + DS$^2$DL(Ours) | ViT/8 | **65.0** | **27.8** | **84.6** | **20.8** | **85.2** | **74.1** |

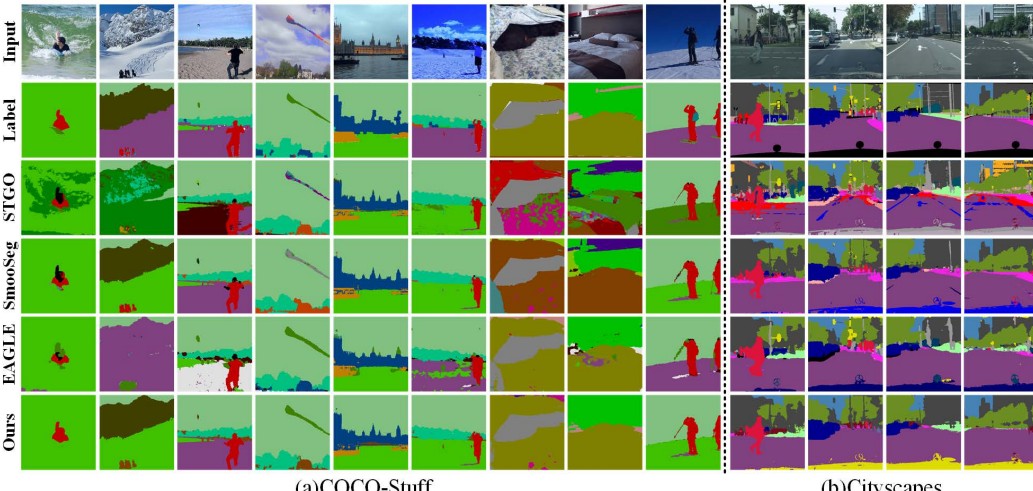

(a)COCO-Stuff        (b)Cityscapes

Figure 3: A qualitative comparison of the (a) COCO-Stuff and (b) Cityscapes, respectively. The comparison includes previous SOTA USS approaches, STEGO, SmooSeg, EAGLE, and ours.

**Qualitative results.** Figure 3 showcases a comparative analysis between our method and several leading segmentation models across multiple benchmarks. Our method consistently outperforms competitors in terms of both segmentation coherence and semantic correctness. While many competitors capture general object boundaries well in simpler scenes, such as those in columns 6 and 8, they often struggle with fine-grained details and semantic precision in more complex regions, such as those in columns 3 and 7. Additionally, STEGO and SmooSeg sometimes result in semantic over-segmentation, such as in column 3 of the right Cityscapes dataset, where structurally distinct but semantically unified components (e.g., lane markings and road) are incorrectly split into separate segments.

**hyperparameter: r** In our framework, the hyperparameter $r$ controls the strength of sample mining by regulating the degree to which hard examples are emphasized during training. When $r = 1$, our DS$^2$DL reduces to the standard binary cross-entropy. As $r$ increases, DS$^2$DL places greater emphasis on hard negative pairs. Figure 4 illustrates the segmentation performance of DS$^2$DL under different values of $r$ across multiple datasets.

According to the reversed-mining formulation, increasing $r$ shifts the emphasis toward easier positive pairs and hard negative pairs. Interestingly, for complex datasets with cluttered scenes and ambiguous regions (e.g., COCO-Stuff and Cityscapes), relatively smaller values of $r$ (e.g., $r = 2.1$

or $r = 2.22$) yield better performance. This is likely because these datasets contain many semantically ambiguous pixels, where assigning higher weights to hard positives helps the model capture subtle intra-class variations. In contrast, for the Potsdam dataset, which features clear boundaries and structured layouts, larger $r$ values (e.g., $r = 6$) prove more effective. In this setting, the abundance of reliable easy positives allows the model to benefit more from emphasizing them along with hard negatives, thereby reinforcing inter-class separation.

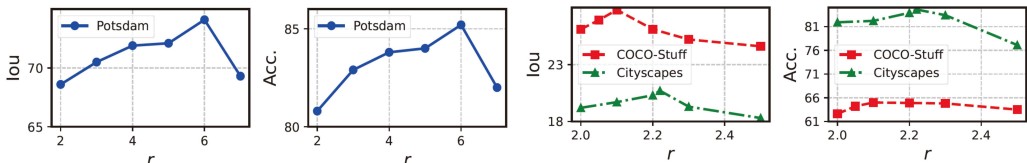

Figure 4: Performance on COCO-Stuff, Cityscapes, and Potsdam datasets under varying values of the hyperparameter $r$. The optimal $r$ varies across datasets, reflecting the ability of our reversed directional mining strategy to adapt to dataset complexity.

These findings suggest that the reversed directional mining strategy adaptively adjusts its focus based on data complexity: smaller values of $r$ are preferable for difficult segmentation tasks, while larger values are more effective in structured and less ambiguous scenarios.

**Ablations.** To further investigate DS$^2$DL, we perform an ablation study on Cityscapes by removing affinity or cluster structure distillation from the objective. As shown in Table 2, we observe a performance drop of 1.6% in Acc and 1.2% in mIoU when the cluster structure distillation is removed. In contrast, removing the affinity structure distillation leads to a much more significant decline, with Acc. decreasing by 52.1% and mIoU by 15.3%. These results underscore the crucial role of affinity structure distillation in guiding the labeling function toward better semantic consistency. While the cluster structure operates in a self-training fashion using pseudo labels, which alone cannot produce sufficiently accurate segmentation maps. The affinity structure inherently enhances the expression of the cluster structure by providing more reliable pairwise relations, suggesting a mutually reinforcing effect between the two. This design choice is further supported by previous works, where affinity relationships have been widely adopted as a means to improve clustering quality and structure discovery Chehreghani (2023); Sadeghi & Armanfard (2024).

Table 2: Ablation results on the Cityscapes dataset.

| Method/Moudle | | Acc. | IOU |
|---|---|---|---|
| DS$^2$DL | | 84.6 | 20.8 |
| Cluster Structure Distillation | w/o orthogonality regularization | 84.3 | 20.5 |
| | w/o pseudo label supervision | 83.8 | 20.1 |
| | w/o Local-to-Global Code Prediction | 83.0 | 19.6 |
| Affinity Structure Distillation | w/o cross-image sample pairs | 55.6 | 12.1 |
| | w/o intra-image sample pairs | 32.5 | 5.5 |

## 5 CONCLUSIONS

We propose a dual-structure self-distilled learning framework for unsupervised semantic segmentation that jointly models pixel-wise affinity and global semantic clustering. The affinity structure is learned via binary classification on pairwise similarities, enabling local semantic relations without explicit classifiers. To mitigate noisy pseudo-labels, we introduce a reversed directional mining strategy that adaptively weights easy and hard samples. The cluster structure arises from soft label features aligned with EMA-maintained global prototypes through a Local-to-Global Code Prediction mechanism, where each pixel predicts its semantic code in the global cluster space. These structures interact via an online self-distillation process, with global prototypes guiding local features. Extensive experiments on COCO-Stuff, Cityscapes, and Potsdam validate the scalability and robustness of our method in producing spatially coherent, semantically meaningful segmentation.

ETHICS STATEMENT

This work focuses on advancing unsupervised semantic segmentation through self-distillation techniques. Our study does not involve human subjects, personally identifiable information, or sensitive data. All datasets used are publicly available benchmark datasets (e.g., COCO-Stuff, Cityscapes, Potsdam) that have established licenses for academic research. We ensured compliance with the respective terms of use.

Our method aims to reduce reliance on costly human annotations, which can lower barriers for research and applications in domains with limited labeling resources. While improved segmentation techniques may be applied broadly, we note the potential for misuse in surveillance or other sensitive contexts. To mitigate this, we release our work solely for academic and scientific purposes, with open code to encourage transparency and reproducibility.

REPRODUCIBILITY STATEMENT

We are committed to ensuring the reproducibility of our work. To this end, we will release the complete source code, training scripts, and configuration files upon publication. All experiments were conducted on publicly available datasets (COCO-Stuff, Cityscapes, and Potsdam) with detailed descriptions of preprocessing steps provided in the paper. Hyperparameters, architectural choices, and optimization settings are reported in full, and random seeds were fixed to reduce variance across runs. We also plan to provide pretrained model checkpoints to facilitate fair comparison and future research.

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

# A APPENDIX

## THE USE OF LARGE LANGUAGE MODELS

The authors acknowledge the use of large language models (LLMs) to assist in polishing the writing. All scientific content, experiments, and conclusions were generated solely by the authors.

## A.1 STRUCURE DISTILLATION

Traditional knowledge distillation typically transfers knowledge from a teacher model to a student by aligning the output logits or soft class probabilities. This form of distillation emphasizes instance-level supervision, where the student is encouraged to mimic the teacher's predictions on each individual sample. While effective in classification tasks, such methods may overlook the relational structure among samples, especially in dense prediction tasks like semantic segmentation or clustering.

We propose Dual-Structure Self-Distilled Learning (DS$^2$DL), a novel self-distillation framework that operates entirely within a single network. DS$^2$DL distills semantic information from label space to guide the feature learning, and from global to local, without relying on any external teacher models. To achieve this, DS$^2$DL distills structural knowledge through two complementary perspectives: affinity structure and cluster structure. As shown in Fig. 5, our approach captures the pairwise relationships (affinities) between feature representations and the global grouping behaviors (clusters) emerging in the data. By preserving and transferring these higher-order structures, DS$^2$DL enables the network to enforce semantic consistency across layers and enhance the organization of features in the representation space. This structure-aware self-distillation proves especially effective in unsupervised settings, where traditional label- or teacher-driven supervision is limited or unavailable.

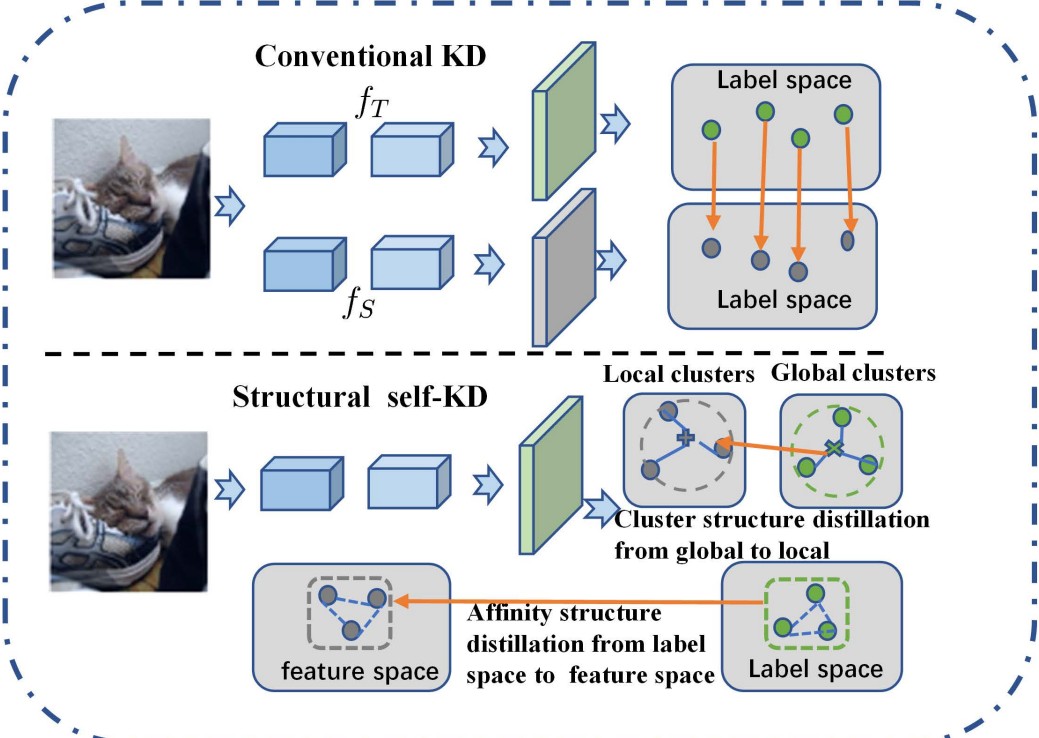

Figure 5: Dual-Structure Self-Distillation.Unlike conventional knowledge distillation (KD), which transfers point-wise outputs from a separate teacher network $f_T$ to a student network $f_S$, our method performs self-distillation within a single network by transferring affinity- and cluster-based structural knowledge from label space to feature space and from global to local, respectively.

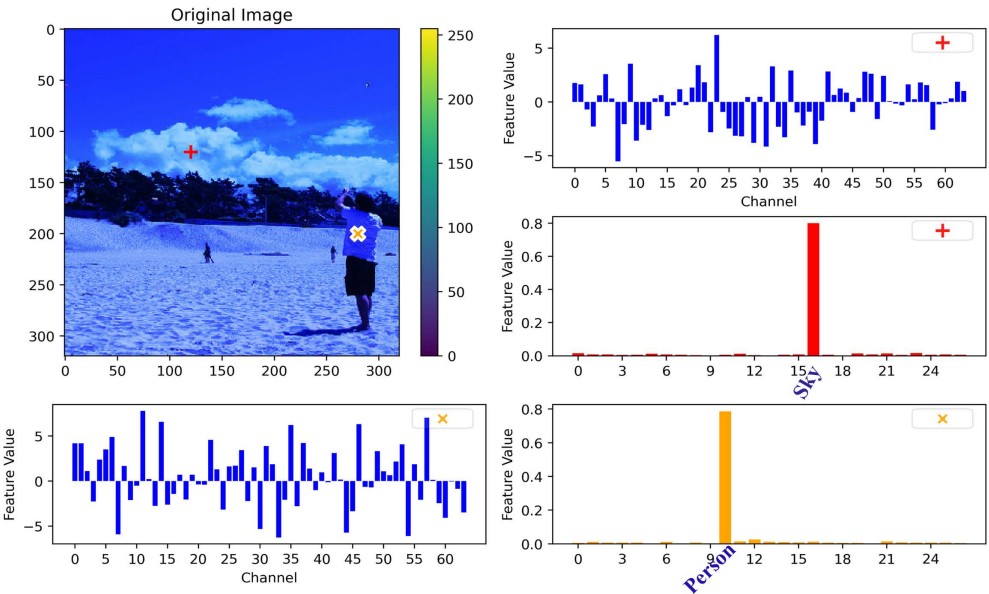

Figure 6: Examples of label features representations of pixels in image. Label feature representations (red and orange) reveal class-specific activations with annotated semantic labels, which are absent in the original features (blue), indicating enhanced semantic separability.

## A.2 LABEL FEATURE

The label feature provides several crucial advantages for unsupervised semantic segmentation, as shown in Fig.6. First, it encodes semantic affinity in a soft and probabilistic manner, rather than relying on discrete and rigid assignments as in hard pseudo-labels Hinton et al. (2015a). This probabilistic representation captures the uncertainty and ambiguity that naturally exist in real-world data. For instance, pixels near object boundaries or in regions with strong occlusion often exhibit mixed semantic cues, and forcing them into a single one-hot label can lead to systematic misclassification. In contrast, soft labels allow the model to represent such cases as weighted combinations of possible classes, leading to more nuanced and flexible decision boundaries.

Second, during training, these soft label features typically evolve into sparse or nearly one-hot-like distributions, which provides a natural mechanism for clustering and semantic grouping across pixels. This property not only makes the optimization more stable by avoiding noisy or contradictory supervision signals, but also helps the model progressively sharpen semantic assignments as training proceeds. In effect, the label features serve both as a form of self-regularization and as an implicit bridge between probabilistic affinity modeling and discrete semantic categorization.

Third, label features are inherently interpretable, since each dimension directly corresponds to a semantic class. This makes it easier to visualize and analyze the learned representations, for example by inspecting class activation patterns, affinity maps, or low-dimensional embeddings of the label space. Moreover, their interpretability facilitates the design of auxiliary objectives or constraints, enabling more effective supervision at the feature level even in the absence of explicit annotations.

Taken together, these properties make label features particularly well-suited for unsupervised semantic segmentation, as they combine the flexibility of probabilistic modeling, the structural advantages of clustering-friendly distributions, and the transparency of class-aligned representations.

## A.3 GRADIENT PERSPECTIVE ON REVERSED DIRECTIONAL MINING

To better understand how reversed directional mining facilitates discriminative learning, we examine the gradients of the two loss components $Q_1(t)$ and $Q_2(t)$ with respect to the similarity score $t =$

$\mathcal{S}(x_i, x_j) + b$. These are defined as:

$$Q_1(t) = \log(1 + \exp(-t/r)), \quad Q_2(t) = \log(1 + \exp(rt)),$$

with corresponding gradients:

$$\frac{\partial Q_1(t)}{\partial t} = -\frac{1}{r} \cdot \sigma\left(-\frac{t}{r}\right), \quad \frac{\partial Q_2(t)}{\partial t} = r \cdot \sigma(rt),$$

where $\sigma(\cdot)$ denotes the sigmoid function. These gradients control how much influence each pair contributes to the optimization based on its similarity score $t$. As shown in Fig. 7.

**Positive Pairs and $Q_1(t)$:** For a positive pair $(x_i, x_j)$, a higher similarity $t$ indicates a more confident prediction. The function $Q_1(t)$ is minimized when $t$ is large, and its gradient $\frac{\partial Q_1}{\partial t}$ is small in this regime. Interestingly, as $t$ decreases (i.e., similarity is low), $\sigma(-t/r)$ increases, and the gradient magnitude grows. However, due to the $1/r$ scaling, the gradient is still relatively modest, and the learning focus remains on positive pairs with already high similarity (i.e., *easy positives*). This reversed emphasis helps reinforce and stabilize well-formed local relationships, while avoiding excessive updates from ambiguous or noisy hard positives.

**Negative Pairs and $Q_2(t)$:** For a negative pair, where ideally $t$ should be small (i.e., low similarity), the loss $Q_2(t)$ grows sharply when $t$ is large, meaning the model incorrectly assigns high similarity to dissimilar pixels. The gradient $\frac{\partial Q_2}{\partial t} = r \cdot \sigma(rt)$ becomes large in this case, resulting in strong repulsive force. This design ensures that *hard negatives*, those with high $t$, are prioritized during training.

**Complementary Effect:** Together, $Q_1(t)$ and $Q_2(t)$ enable the model to concentrate its optimization efforts on **easy positives** (with already high similarity) and **hard negatives** (with confusingly high similarity). This strategy avoids the instability often caused by *hard positive* pairs with noisy or ambiguous labels and promotes confident and reliable updates. The resulting gradient profile leads to more stable convergence and enhanced intra-class compactness and inter-class separation.

### A.4 THE IMPACT OF CRF

Conditional Random Fields (CRFs) are widely used in semantic segmentation to refine object boundaries and recover fine-grained details Krähenbühl & Koltun (2011). By explicitly modeling both pairwise semantic relations and global semantic grouping at the feature level, our approach promotes semantic coherence and structural consistency across the image. While CRF can be optionally applied to further sharpen boundary details by leveraging low-level appearance cues, it only provides marginal improvements. This demonstrates that our dual-structure distillation alone is sufficient for generating high-quality segmentation maps, and the CRF module serves merely as a lightweight refinement step rather than a core component.

As shown in Table 3, our DS$^2$DL based on affinity structure and cluster structure distillation already achieves strong segmentation performance without relying on CRF post-processing. Our method achieves superior mIoU compared to both Smooseg and STEGO, even without relying on CRF post-processing techniques, demonstrating the strength of our structural learning framework.

Table 3: Experimental results of the impact of CRF on SmooSeg, STEGO, and DS$^2$DL.

| | COCOStuff | | Cityscapes | | Potsdam-3 | | Avg. | |
| --- | --- | --- | --- | --- | --- | --- | --- | --- |
| | Acc | mIoU | Acc | mIoU | Acc | mIoU | Acc | mIoU |
| STEGO w/o CRF | 46.5 | 22.4 | 63.5 | 16.8 | 74.1 | 58.9 | 61.4 | 32.7 |
| STEGO w CRF | 48.3 | 24.5 | 69.8 | 17.6 | 77.0 | 62.6 | 65.0 (+3.6) | 34.9 (+2.2) |
| SmooSeg w/o CRF | 60.6 | 25.2 | 79.8 | 18.0 | 81.4 | 68.4 | 73.9 | 37.2 |
| SmooSeg w CRF | 63.2 | 26.7 | 82.8 | 18.4 | 82.7 | 70.3 | 76.2 (+2.3) | 38.5 (+1.3) |
| DS$^2$DL w/o CRF | 62.0 | 26.0 | 80.2 | 19.6 | 83.6 | 71.8 | 75.3 | 39.1 |
| DS$^2$DL w CRF | 65.0 | 27.8 | 84.6 | 20.8 | 85.2 | 74.1 | 78.3 (+3.0) | 40.9 (+1.5) |

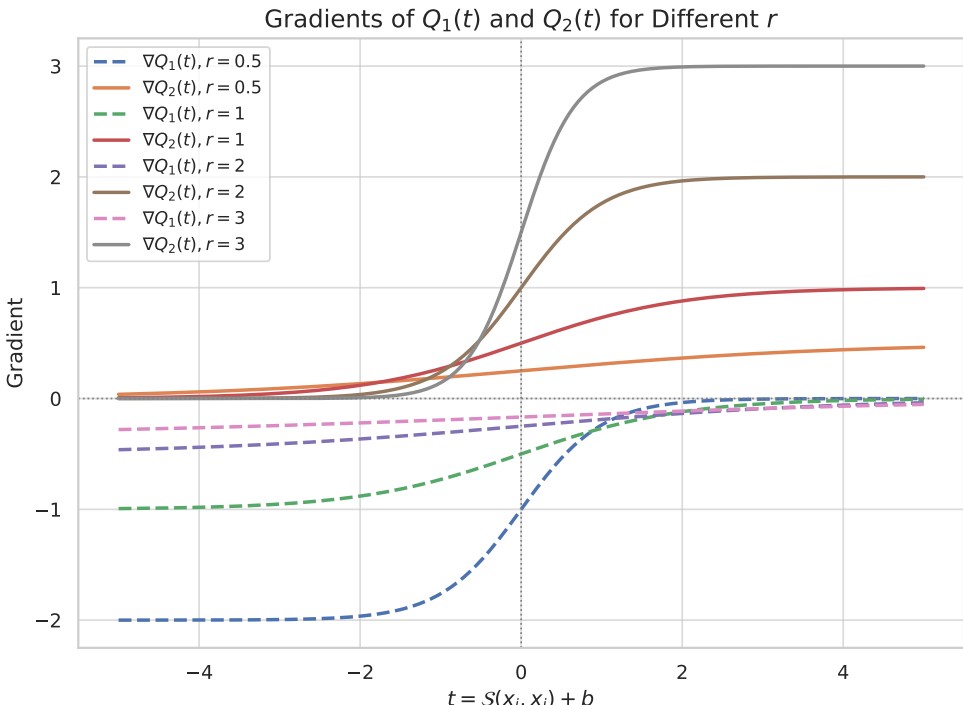

Figure 7: Gradient curves of $\nabla Q_1(t)$ and $\nabla Q_2(t)$ under different $r$ values. Larger $r$ leads to sharper emphasis on different pairs, enabling reversed directional mining to focus learning on easy positive and hard negative pairs.

## A.5   EXTENDED RESULTS AND DISCUSSIONS

To further validate the effectiveness and robustness of our proposed self-distilled dual-structure learning framework, we conduct additional experiments including extended qualitative visualizations and hyperparameter configurations. These results consistently demonstrate that our approach not only generalizes well across diverse segmentation benchmarks but also maintains strong semantic coherence even without CRF post-processing. The observed improvements confirm the complementary roles of local affinity modeling and global cluster alignment, reinforcing the benefits of dual-structure learning in fully unsupervised settings.

**Hyperparameter configurations**   We analyze the influence of the temperature parameter $\tau$ on the performance of DS$^2$DL using the Cityscapes dataset, with results presented in Fig. 8. In theory, a smaller $\tau$ sharpens the softmax distribution, thereby amplifying the training signal by producing stronger gradients. $\tau$ has a pronounced impact on model performance. DS$^2$DL performs well when $\tau \leq 0.1$, while its effectiveness deteriorates significantly for $\tau \geq 0.2$, where the softmax output becomes overly smooth and less informative. These observations highlight the importance of carefully tuning $\tau \leq 0.1$ to preserve meaningful contrastive supervision during training.

The hyperparameter $\varepsilon$ controls the strength of the entropy regularization term $\mathcal{H}(\mathbf{Q})$ in the objective function, which encourages the distribution $\mathbf{Q}$ to be more uniform and thus prevents overly confident or sparse assignments. We investigate the impact of the entropy regularization parameter $\varepsilon$ on the performance of DS$^2$DL on the Cityscapes dataset, and report the results in Fig. 9. As can be seen, when $\varepsilon$ is close to 0.1, this regularization effectively smooths the optimization landscape, providing sufficient exploration while maintaining meaningful gradients from the data fitting term $\mathrm{Tr}(\mathbf{Q}^\top \mathbf{C}^\top \mathbf{X})$. This balance leads to improved stability and generalization in training. However, as $\varepsilon$ increases beyond 0.2, the entropy term dominates, causing the distribution $\mathbf{Q}$ to become too uniform and the model to underfit the data, which results in degraded performance. Conversely, if $\varepsilon$ is too small, the model may produce overly sharp or unstable solutions. Therefore, tuning $\varepsilon$

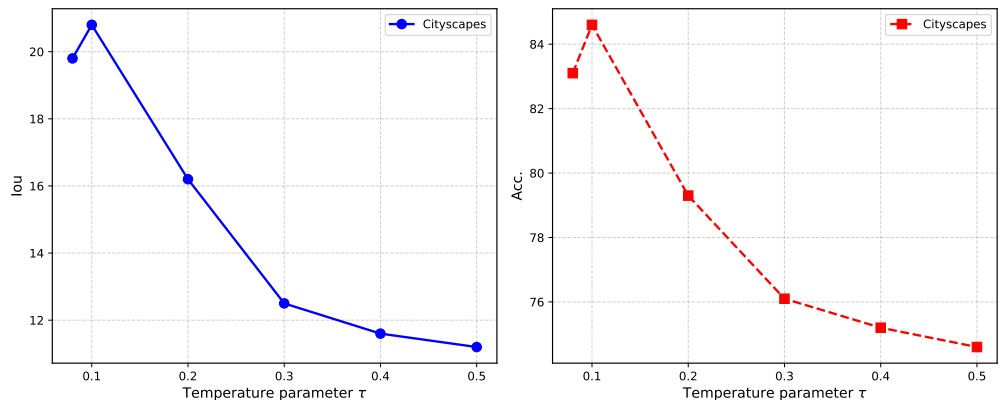

Figure 8: Sensitivity study on the temperature parameter $\tau$.

is critical, and our experiments suggest that a value near 0.1 provides the best trade-off between regularization and data fitting.

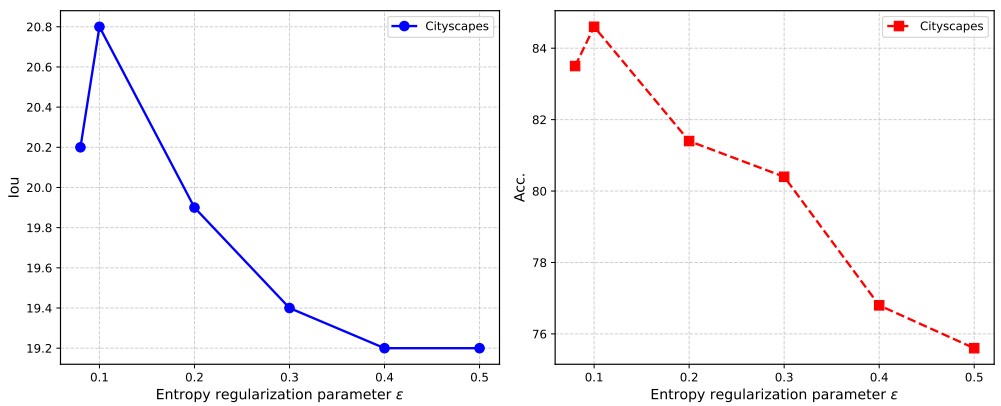

Figure 9: Sensitivity study on the entropy regularization parameter $\varepsilon$.

**Qualitative visualizations**   We provide qualitative segmentation results with and without CRF post-processing in Fig. 10 for the COCO-Stuff dataset, Fig. 11 for the Cityscapes dataset, and Fig. 12 for the Potsdam-3 dataset. Overall, our DS$^2$DL method consistently yields more accurate and spatially coherent segmentation maps compared to SmooSeg. While CRF contributes to refining fine-grained details for both DS$^2$DL and SmooSeg. These results highlight the inherent strength of our self-distilled dual-structure learning framework in modeling both local affinity and global cluster structures for improved unsupervised segmentation.

We present several challenging examples from the COCOStuff dataset in Figure 13 to compare the predictions of SmooSeg and DS$^2$DL. While both methods encounter difficulties in accurately segmenting these complex scenes, DS$^2$DL consistently produces more semantically coherent results. This demonstrates the effectiveness of our proposed affinity and cluster structure distillation in improving unsupervised semantic segmentation, particularly in challenging scenarios.

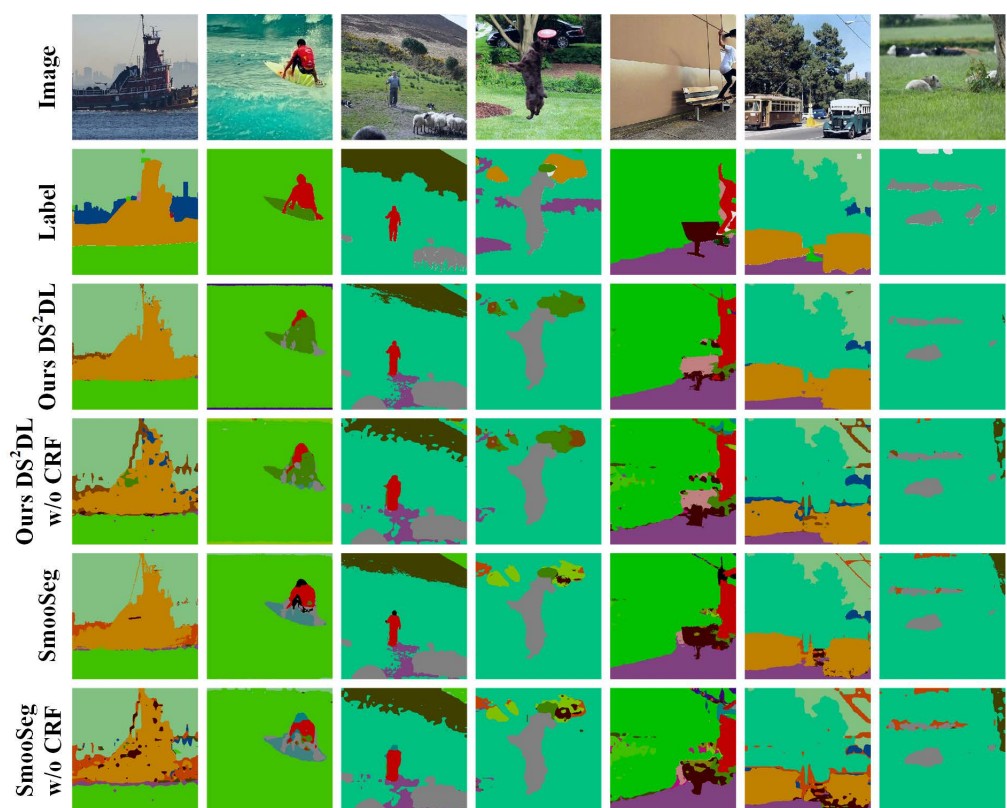

Figure 10: Qualitative results of SmooSeg and DS$^2$DL with and without CRF on the COCOStuff dataset.

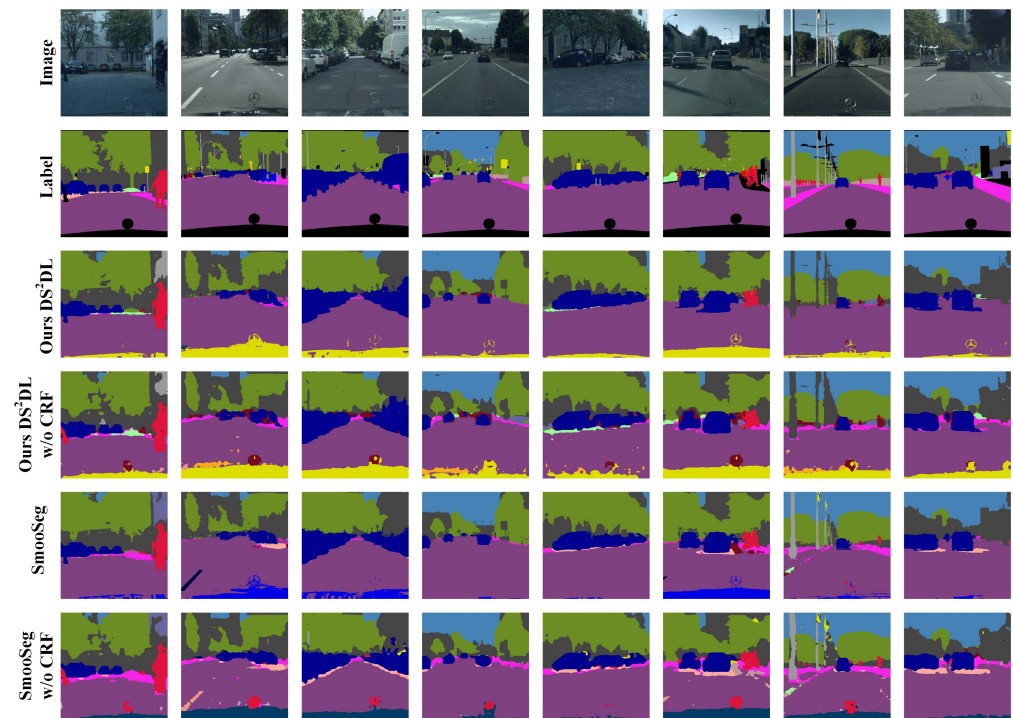

Figure 11: Qualitative results of SmooSeg and DS$^2$DL with and without CRF on the Cityscapes dataset.

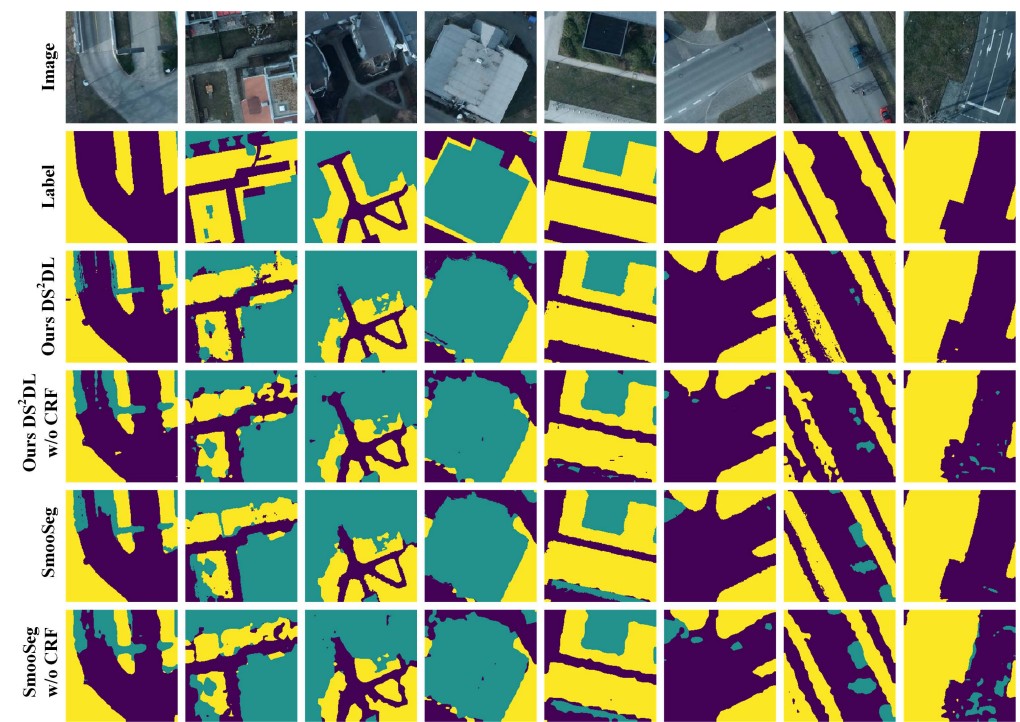

Figure 12: Qualitative results of SmooSeg and DS$^2$DL with and without CRF on the Potsdam-3 dataset.

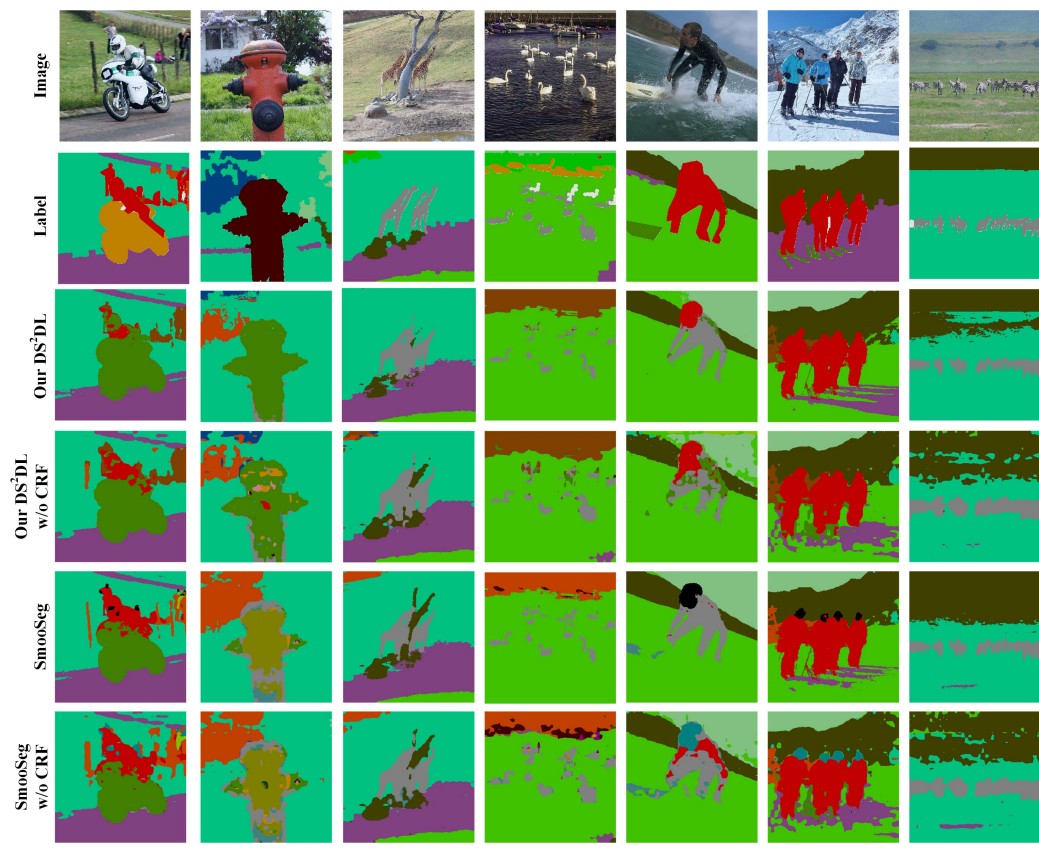

Figure 13: Difficult examples predicted by SmooSeg and DS$^2$DL on the COCOStuff dataset.

