# OpenReview forum: "Dual-Structure Self-Distilled Learning for Enhancing Unsupervised Semantic Segmentation"
_ICLR.cc/2026/Conference — ICLR 2026 Conference Desk Rejected Submission_

### Official Review · Reviewer_e8JZ · 2025-10-28

**Soundness:** 3
**Presentation:** 2
**Contribution:** 1
**Rating:** 4
**Confidence:** 4

**Summary:**

This paper introduces Dual-Structure Self-Distilled Learning (DS²DL), a new framework for unsupervised semantic segmentation (USS) that aims to jointly model local pixel affinities and global clustering structures. The core idea is a dual self-distillation scheme operating within a single network. An "Affinity Structure Distillation" module reframes segmentation as a binary classification of pixel pairs, guided by soft labels derived from the model's own predictions. Complementing this, a "Cluster-Level Structure Distillation" module enforces global consistency by aligning local pixel predictions with global semantic prototypes updated via Exponential Moving Average (EMA). The authors claim substantial improvements over strong baselines like STEGO on COCO-Stuff, Cityscapes, and Potsdam-3 datasets.

**Strengths:**

1. The method achieves state-of-the-art or highly competitive performance across three standard benchmarks.
2. The paper includes a detailed "Implementation Details" section (4.3) that specifies architectures, optimizers for different datasets, momentum coefficients, and other crucial hyperparameters.
3. The overall organization of this paper is easy to follow.

**Weaknesses:**

1. The central novelty of this paper is largely derivative of prior work:
- Learning pairwise affinities is a classic approach in clustering. Formulating it as a binary classification task is not new (e.g., seen in deep metric learning). The use of cosine similarity from label features as pseudo-ground truth is a straightforward extension of self-training paradigms.
- The "Cluster-Level Structure Distillation," involving local-to-global code prediction and EMA-updated global prototypes, is heavily inspired by SwAV (Caron et al., 2020). The paper appropriately cites SwAV but downplays the extent of the methodological borrowing, presenting it as a novel component of its framework.
- EMA-update is the cornerstone of many recent self-supervised and semi-supervised methods like DINO and BYOL. The paper fails to differentiate its use of self-distillation from this extensive body of work in a meaningful way.

2. The proposed "Reversed Directional Mining" strategy (Section 3.3) is counter-intuitive and inadequately justified：
- Hard example mining literature almost universally argues for focusing on difficult samples to improve decision boundaries. Why this paper considers it is beneficial to focus on easy positives? The authors provide no theoretical or strong empirical evidence to support this reversed logic.
- The ablation study in Table 2 is insufficient. Removing positive or negative sample pairs entirely is too coarse. A more convincing study would compare the proposed reversed mining to standard hard-mining and no mining (r=1) to isolate the actual effect of this specific strategy.
- The performance graphs in Figure 4 show that the optimal $r$ varies wildly across datasets, suggesting this is a sensitive hyperparameter requiring careful tuning rather than a robust, generalizable mechanism.

3. The ablation study presented in Table 2 fails to convincingly demonstrate the necessity of core components:
- Removing "affinity structure distillation" causes a massive performance drop (Acc. from 84.6 to 32.5). This is unsurprising, as it appears to remove the primary loss function driving local consistency. This doesn't prove the superiority of the proposed affinity module, but rather that local consistency learning is essential.
- Removing "cluster structure distillation" results in a much smaller drop (Acc. from 84.6 to 83.0). This raises questions about the importance of this entire branch of the model. Are the performance gains primarily from the affinity learning part?
- The study does not ablate the cross-image pixel pairing, the Sinkhorn algorithm for optimal transport, or the orthogonality regularization. The contributions of these individual design choices remain unverified.

**Questions:**

* Removing "cluster structure distillation" results in a much smaller drop (Acc. from 84.6 to 83.0). This raises questions about the importance of this entire branch of the model. Are the performance gains primarily from the affinity learning part?
* Hard example mining literature almost universally argues for focusing on difficult samples to improve decision boundaries. Why this paper considers it is beneficial to focus on easy positives?

---

> ### Author Response · Authors · 2025-11-26
> **Official Comment by Authors**
>
> We thank the reviewer for devoting time to reviewing our paper, and for their encouraging comments and insightful questions, below we respond to each of the points in turn.
>
> ---
>
> **Q1:** “The central novelty of this paper is largely derivative of prior work".
>
> **A1:**  We appreciate the opportunity to clarify the novel contributions of our work and its distinctions from prior studies. While some components in our method are inspired by existing works, the core novelty lies in **how these techniques are adapted, combined, and extended for the specific challenges of unsupervised semantic segmentation (USS)**. Unlike standard classification, USS lacks both image-level and pixel-level labels, leaving only **pixel-wise relationships** to exploit. In this context, architectural choices alone are often insufficient, and the key innovation lies in the **learning strategy for capturing local and global semantic structures under fully unsupervised constraints**.
>
>
> (1) Regarding pairwise affinity learning and binary classification, although similar ideas exist in previous literature, our contribution is **not in the use of pairwise affinities per se**, but in how pixel-level supervision is constructed, optimized, and combined with cluster-level guidance. Our pseudo-labels are dynamically generated from evolving label-feature embeddings, enabling correction of noisy affinities, unlike conventional static pseudo-labels used in metric learning or self-training. Furthermore, we introduce a **parametric pairwise objective** with gradient shaping, which downweights easy positives and emphasizes hard negatives to stabilize cluster boundaries under noisy supervision. In addition, the cosine similarity between label features is used as a **cluster-consistent affinity signal** that incorporates global semantics rather than as a simple per-sample pseudo-label heuristic. These design choices, validated empirically across multiple datasets, yield significant performance improvements beyond standard methods.
>
>
> (2) For Cluster-Level Structure Distillation (CLSD), although SwAV inspires the use of prototype-based prediction, our approach differs substantially in **objective, mechanism, and learning dynamics**. While SwAV focuses on instance-level self-supervision to learn argmuented visual features, CLSD aligns local pixel-cluster assignments with global semantic prototypes, realizing **cluster-level structure transfer**, a distillation objective absent in SwAV. SwAV relies on batch-level OT assignment as its core mechanism, whereas our L2G module employs EMA-updated global prototypes and local-to-global code prediction, with OT used only for refining cluster-level codes rather than as the main optimization target. Unlike SwAV, which clusters local features into prototypes, CLSD distills global cluster distributions into local features, enabling local predictions to be guided by global semantic structures and achieving **bidirectional local-to-global semantic alignment**.
>
>
> (3) Regarding EMA, our use is fundamentally different from DINO and BYOL. In DINO/BYOL, EMA maintains a momentum teacher encoder to provide stable instance-level contrastive or self-distillation targets. In contrast, we **update only the global cluster prototypes via EMA**, serving as a stabilizer for the dataset-level semantic distribution. The core learning signal comes from **Local-to-Global (L2G) code prediction**, which enforces patch-level cluster codes to align with EMA-stabilized global prototypes, a mechanism absent in DINO/BYOL and other teacher-based methods.
>
>
> In summary, the innovation of this work lies not merely in the individual components, but in **the integration of cluster-aware pairwise learning**, **Reversed Directional Mining (RDM)**, __Local-to-Global code prediction (LGP)__, and **EMA-stabilized global prototypes specifically for USS**. This combination, together with task-specific adaptations, goes beyond simple adaptation or borrowing and demonstrates significant empirical gains over strong baselines, validating its methodological originality and practical impact.

---

> ### Author Response · Authors · 2025-11-26
> **Official Comment by Authors**
>
> **Q2:** “The proposed "Reversed Directional Mining" strategy (Section 3.3) is counter-intuitive and inadequately justified.”
>
> **A2:** Thank you for the insightful comments regarding the Reversed Directional Mining (RDM) strategy in Section 3.3. We appreciate the opportunity to clarify both the theoretical motivation and empirical validation of this approach.
>
> (1) Classical hard example mining typically assumes clean and reliable labels, where emphasizing hard positives near semantic boundaries improves decision boundaries. In contrast, in **unsupervised pixel affinity learning**, positive labels are derived from soft and noisy pseudo-labels. We observe that “hard positives” (low-similarity pairs marked as positive) overwhelmingly correspond to boundary pixels, cross-class mixtures, or other noisy assignments, which makes them unreliable supervisory signals. Emphasizing these pairs can lead to unstable training in the unsupervised semantic segmentation task. Our RDM strategy is designed to address this issue by **down-weighting hard positives while emphasizing hard negatives**, which remain informative. Analytically, the gradient behavior of the hyperparameter $r$ (Appendix A.3) demonstrates how this adaptive weighting stabilizes pairwise supervision under noisy conditions.
>
> (2) We have provided a direct empirical comparison between RDM and standard BCE ($r=1$) on Cityscapes to isolate the effect of the reversed mining strategy. When RDM is replaced with standard BCE, performance drops significantly from 84.6 → 76.1 Acc and 20.8 → 17.92 mIoU, this confirms that attenuating easy positives while highlighting hard negatives provides more stable and effective supervision than standard hard mining. This result complements the ablation study in Table 2 and demonstrates that RDM is not counter-intuitive but **tailored for the noise characteristics of unsupervised segmentation**.
>
> | Setting / Loss | Acc (%) | mIoU (%) |
> |---|---:|---:|
> | RDM ($r=2.22$) | 84.6 | 20.8 |
> | Standard BCE ($r = 1$) | 76.1 | 17.92  |
>
> (3) Regarding the hyperparameter $r$, we acknowledge that its optimal value varies across datasets, as illustrated in Figure 4. This variability reflects adaptation to dataset-specific characteristics rather than a flaw: different datasets exhibit varying degrees of intra-class similarity and inter-class separation. For example, cluttered datasets like COCO-Stuff and Cityscapes benefit from focusing on easy positives to capture fine-grained distinctions, whereas datasets with cleaner boundaries, such as Potsdam-3, benefit from emphasizing hard negatives (larger $r$ values). Therefore, the sensitivity of $r$ is an expected property that allows RDM to flexibly handle diverse scene complexities.
>
> Empirically, once an appropriate $r$ is chosen, RDM generalizes robustly across datasets, demonstrating both stability and practical applicability.
>
> In summary, RDM is a **task-specific mechanism** designed to stabilize pixel-level affinity learning in the presence of noisy pseudo-labels. Theoretical reasoning, detailed ablations, and cross-dataset experiments together substantiate its effectiveness, while its adjustable hyperparameter $r$ provides flexibility to adapt to varying scene complexities.

---

> > ### Author Response · Authors · 2025-11-26
> > **Official Comment by Authors**
> >
> > **Q3:** “The ablation study presented in Table 2 fails to convincingly demonstrate the necessity of core components.”
> >
> > **A3:** We thank the reviewer for the insightful comments regarding Table 2 and the necessity of core components. We provide a detailed response addressing each concern.
> >
> > (1) Regarding the large performance drop when removing the **affinity structure distillation**, we agree that it highlights the essential role of local consistency learning in unsupervised semantic segmentation (USS). In dense prediction tasks, pixel–pixel relations carry the primary semantic structure, and the affinity branch naturally contributes most to performance. Importantly, disabling the affinity branch does not merely remove a generic smoothness term; it eliminates a mechanism that derives pairwise supervision from EMA-updated label-feature embeddings, integrates cross-image pairing, and applies reversed directional mining to modulate gradients for positives and negatives. The ablated model thus reduces to a self-labeling baseline with CRF post-processing, akin to STEGO/SmooSeg. The comparison with these strong baselines in Table 1 demonstrates that our affinity module provides substantial gains beyond generic local consistency, which we will clarify more explicitly in the revised manuscript. We will also update Table 2’s description to emphasize that this ablation answers: “Is the proposed affinity distillation necessary beyond standard self-labeling + CRF?”, with the large performance gap confirming its necessity.
> >
> > (2) Concerning the smaller drop when removing the **cluster structure distillation**, this behavior aligns with the design intent. The affinity branch drives the majority of pixel-level local consistency, whereas the cluster branch acts as a complementary global regularizer through Local-to-Global (L2G) code prediction. Despite a smaller numerical drop (Acc. 84.6 → 83.0), the cluster branch consistently improves metrics across datasets (Acc and mIoU gains, see Table 2 and Appendix), enhances global coherence, and stabilizes rare-class predictions (Fig. 3). Thus, while the primary gains originate from affinity learning, the cluster structure contributes meaningful and complementary improvements.
> >
> > (3) Regarding the lack of ablations for **cross-image pixel pairing, Sinkhorn OT, and orthogonality regularization**, we clarify as follows:
> >
> > (i)	**Orthogonality regularization**: Table 2 includes the ablation “w/o orthogonality regularization,” showing a small but consistent drop (Acc 84.6 → 84.3, mIoU 20.8 → 20.5), validating its contribution in preventing prototype collapse and encouraging balanced code usage.
> >
> > (ii)	**Cross-image pixel pairing**: In USS, explicit positive/negative labels are absent, so cross-image pairs serve as valid negative proxies. Ablating cross-image negative pairs leads to severe degradation (Acc 83.0 → 55.6, mIoU 19.6 → 12.1), confirming their necessity. To avoid ambiguity, we will revise Table 2 labels: “w/o negative sample pairs” → “w/o cross-image sample pairs” and “w/o positive sample pairs” → “w/o intra-image sample pairs.”
> >
> > (iii)	**Sinkhorn OT**: OT is an inseparable part of the L2G structure, enabling label-group alignment that differentiates L2G from naive pseudo-label supervision. Ablating OT is equivalent to disabling L2G itself. We will clarify this in the revised Table 2 (“w/o OT (L2G disabled)”) and emphasize that OT and L2G form a coupled structural unit, making separate ablation redundant.
> >
> > In summary, the ablation study comprehensively demonstrates the necessity and contributions of each component. The affinity branch is the dominant driver of local consistency, the cluster branch provides complementary global regularization, and cross-image pairing, OT, and orthogonality regularization are essential to stabilize learning and improve segmentation quality. We will revise the manuscript to clarify these points and provide updated labels and explanations in Table 2 to ensure the experimental design is fully interpretable and convincing.

---

### Official Review · Reviewer_Tm7S · 2025-10-30

**Soundness:** 3
**Presentation:** 3
**Contribution:** 3
**Rating:** 8
**Confidence:** 4

**Summary:**

The paper introduces Dual-Structure Self-Distilled Learning, a novel framework for Unsupervised Semantic Segmentation (USS). The authors argue that existing USS methods struggle to effectively capture semantic structures at different levels of abstraction simultaneously. Specifically, most methods do not explicitly and jointly model both the affinity structure (fine-grained, local pixel relationships) and the cluster structure (high-level, global semantic groups). The proposed DS^2DL framework addresses this gap by using a dual self-distillation scheme: Affinity Structure Distillation and Cluster Structure Distillation, within a single network. It works by enforcing semantic consistency at both the local and global levels simultaneously, transferring semantic cues from the label space into the feature space. The method demonstrates significant performance gains over the strong baseline STEGO and other state-of-the-art methods on three benchmark datasets: COCO-Stuff, Cityscapes, and Potsdam-3.

**Strengths:**

**Originality:** The paper's primary originality lies in its holistic framework design. It is the first to explicitly and jointly model the affinity (local) and cluster (global) structures for unsupervised semantic segmentation within a unified self-distillation framework. While prior works have used affinity or clustering, the synthesis of these two as a dual self-distillation mechanism is novel. Key original contributions include Dual-Structure Self-Distillation and Reversed Directional Mining.

**Quality:** The paper is of high quality. The methodology is well-reasoned, and the empirical evaluation is robust.

**Clarity:** The paper is well-written, well-structured, and effectively uses visuals to explain its core concepts.

**Significance:** This work makes a significant contribution to the challenging field of unsupervised semantic segmentation.The method achieves substantial improvements over a wide range of state-of-the-art baselines, including the strong STEGO model. The reported gains (e.g., +16.7 Acc on COCO-Stuff, +14.8 Acc on Cityscapes and +11.5 mIoU on Potsdam-3 over STEGO) are significant.

**Weaknesses:**

- The caption of Figure 1 is too brief. It would benefit from a more detailed introduction and explanation of the overall framework. Additionally, please clarify the meaning of the icon between the "weight sharing" components—is it representing the visual features? The lightweight architecture, which consists of a linear layer and a two-layer MLP with SiLU activations, should also be explicitly illustrated in the figure for completeness.
- As Reversed Directional Mining (RDM) is a key novel component. its experimental validation should be more comprehensive. Although the paper analyzes the hyperparameter $r$ (Figure 4) and notes that $r=1$ corresponds to standard binary cross-entropy (BCE), the main ablation study (Table 2) does not include a direct comparison between the full model with RDM (e.g., $r=2.22$ on Cityscapes) and the baseline model using standard BCE ($r=1$). Including this comparison would more clearly demonstrate the effectiveness of RDM.
- The Quantitative Evaluation section requires substantial improvement. Beyond listing numerical comparisons, the authors should summarize the overall trends observed in the results and provide insightful analysis explaining the performance differences.

**Questions:**

- Would it be useful to introduce hyperparameters in Eq (8) to create a weighted loss function for model training?

---

> ### Author Response · Authors · 2025-11-26
> **Official Comment by Authors**
>
> We thank reviewer Tm7S for their encouraging and insightful comments. We would respond to each of the concerns and questions in turn:
>
> ---
>
> **Q1:** “Caption of Figure 1 is too brief; needs clearer explanation and clarification of icons and lightweight architecture.”
>
> **A1:**  We have revised the caption of Figure 1 to provide a clearer and more detailed explanation, covering the backbone, the lightweight projector, and the roles of the two distillation branches. We explicitly clarify that the two branches share the weights of the linear projection layer rather than visual features, and the caption now details the function of each major component.
>
> The revised caption in the manuscript reads:
>
> Figure 1: Overview of our DS\$^2$DL framework. Our dual-structure framework consists of two separate branches: Cluster-Level Structure Distillation (CLSD) and Affinity Structure Distillation (ASD). ASD distills semantic relationships from learned label features into pixel-level representations to promote local semantic consistency, while CLSD adopts a Local-to-Global Code Prediction strategy to capture global semantic grouping. In this framework, DINO-pretrained features are duplicated and mapped into the same feature space through a shared linear projection layer. One projected feature is further processed by a two-layer MLP for cluster structure distillation, and the other is used for affinity structure distillation.
>
> The updated figure reflecting these structural details has been included in the revised manuscript.
>
>
> ---
>
> **Q2:** “Requests a direct ablation comparing RDM (e.g., $ r>1$) with standard BCE ($r=1$) to validate the effectiveness of RDM.”
>
> **A2:** We agree that a direct comparison between RDM and standard BCE is important to highlight the contribution of RDM. As noted in Section 3.3, RDM reduces to standard BCE when $r=1$. In the revised manuscript, we include this comparison.
>
> On Cityscapes, replacing RDM with standard BCE ($r=1$) reduces performance from 84.6%→ 76.1% Acc and 20.8% → 17.92% mIoU. This result confirms that RDM, by emphasizing easy positives and hard negatives, provides more stable and effective pairwise supervision than standard BCE, leading to significant performance improvements.
>
> | Setting / Loss | Acc (%) | mIoU (%) |
> |---|---:|---:|
> | RDM ($r=2.22$) | 84.6 | 20.8 |
> | Standard BCE ($r = 1$) | 76.1 | 17.92  |
>
> These additional ablation results clearly demonstrate the effectiveness of RDM and substantiate its role as a key component in our framework.

---

> ### Author Response · Authors · 2025-11-26
> **Official Comment by Authors**
>
> **Q3:** “Requests a clearer summary and analysis of the quantitative results beyond listing numbers.”
>
> **A3:**  We have expanded the Quantitative Evaluation section to not only report numerical results but also summarize the overall trends and provide insights into the observed performance differences. Specifically, our method achieves notable gains over the strong baseline STEGO on multiple datasets: COCO-Stuff (+16.7 Acc, +3.3 mIoU), Cityscapes (+14.8 Acc, +3.2 mIoU), and Potsdam-3 (+8.2 Acc, +11.5 mIoU).
>
> These improvements can be attributed to the proposed dual-structure self-distilled framework. The affinity structure models local pixel-wise interactions via binary classification of pairwise similarities and is enhanced with the Reversed Directional Mining (RDM) strategy, which improves spatial coherence and stabilizes pairwise supervision. Simultaneously, the cluster structure leverages a Local-to-Global code Prediction (LGP) mechanism to align pixel-level features with EMA-updated global prototypes, providing consistent global semantic guidance.
>
> Together, these two complementary structures enable our model to capture both local relationships and global semantic context, leading to more accurate, structured, and consistent segmentation. The expanded analysis in the revised manuscript highlights these trends and explains the performance differences observed across datasets.
>
> ---
>
> **Q4:** “Would it be useful to introduce hyperparameters in Eq.(8) to create a weighted loss function for model training?”
>
> **A4:** In our framework, the loss terms are naturally organized into two tightly coupled groups: the __affinity-structure distillation loss__  $ L_{\text{p}}$ and the  __cluster-structure regularization loss__  $L_{\text{clus}} = L_{L2G} + L_{\text{orth}} + L_{\text{data}}$. These two components are jointly optimized to reinforce each other.
> To evaluate the potential benefit of introducing a weighting hyperparameter, we consider a weighted overall objective:
> $L_{\text{total}} = L_{\text{aff}} + \lambda L_{\text{clus}} $,
>
> We conducted experiments on Cityscapes with $\lambda$ ranging from 0.2 to 5. As summarized in the table below, both mIoU and Accuracy remain highly stable across this range. The mIoU varies by only 0.2%, and Accuracy by 1.65%, indicating that our method is not sensitive to the precise loss weighting.
>
>
> | $\lambda$ | Accuracy | mIoU |
> |-------|----------|------|
> | 0.2   | 82.9649  | 20.5523 |
> | 0.4   | 83.0437  | 20.5909 |
> | 0.6   | 83.3610  | 20.6151 |
> | 0.8   | 84.0609  | 20.7147 |
> | 1.0   | **84.6127** | **20.7530**  |
> | 1.2   | 84.0609  | 20.7151 |
> | 1.6   | 84.0611  | 20.7149 |
> | 2.0   | 84.0610  | 20.7148 |
> | 5.0   | 84.0643  | 20.7183 |
>
> These results indicate that our current unweighted formulation already achieves stable and robust training. Introducing an additional weighting hyperparameter does not yield significant improvement and is therefore not critical.

---

### Official Review · Reviewer_x2DC · 2025-10-31

**Soundness:** 2
**Presentation:** 2
**Contribution:** 2
**Rating:** 2
**Confidence:** 4

**Summary:**

This study aims to improve the feature representation in unsupervised semantic segmentation. The network first projects the features into a lower-dimensional clustered label space and then use the similarity in the label space to supervise the feature similarity for feature clustering. It then utilize a cluster structure to derive semantic codes from prototypes and align pixel prediction using optimal transport theory. It is verified on different benchmarks to reach the state-of-the-art performance.

**Strengths:**

- The paper is mostly easy to follow, and the ideas are presented clearly. Starting from the inspiration from self-distillation and cross-modal distillation, the authors smoothly present the idea to utilize the distillation from affinity and clustering structures to make the feature space more compact.
- The experimental performance is strong on benchmarks like COCO-Stuff, Cityscapes, and Potsdam-3 with its Acc and mIOU reaching state-of-the-art compared to previous literature.
- The ablation over each component is considered necessary and compelling. The advantage of cluster sturcture distillation over affinity structure distillation is surprising.

**Weaknesses:**

1. Originality issues. The two major components--affinity structure learning and cluster structure distillation, are not designed and invented in this study.  Projecting the features into a lower-dimensional clustered label space and then supervising the feature similarity using the similarity in the lower space has been a common practice in unsupervised learning. Simply assembling these components into unsupervised semantic segmenation is not considered to reach the standard of novelty of this conference.
2. What really concerns me is that the authors failed to acknowledge and discuss the differences with these classic works. For example, using binary classification as the cluster supervision loss as in Eq. 1 had been widely utilized in [1] and many following papers. The online coding using optimal transport theory as in Eq. 5 has long been proposed in SwAV[2]. Both of these studies are not properly referred to and discussed with in this manuscript, which leaves a compression that these methods are originated in this work.
2. The sampling mining function may require further proof. The authors propose to emphasize more on the negative pair loss and decrease the importance of positive pairs. But the intuition is not explicitly explained in Sec. 3.3 and other part of the manuscript.
3. As the authors emphasize on the soft label similarity learning instead of using hard labels in Eq. 1, there should also be an ablation on the advantage of such a change compared to common practice as well.

[1] Wen, Yandong, et al. "Pairwise similarity learning is simple." Proceedings of the IEEE/CVF International Conference on Computer Vision. 2023.
[2] Caron, Mathilde, et al. "Unsupervised learning of visual features by contrasting cluster assignments." Advances in neural information processing systems 33 (2020): 9912-9924.

**Questions:**

The originality issue is a great concern in this study and I would like to hear from the authors' rebuttal.

---

> ### Author Response · Authors · 2025-11-26
> **Official Comment by Authors**
>
> Thank you for the careful and constructive comments in this evaluation effort. We first clarify our contributions and how they differ from prior work, and then respond to each concern point-by-point.
>
> ---
>
> **Q1:** “On originality and connection to prior work”
>
> **A1:** While projecting features into a lower-dimensional clustered label space and supervising feature similarity is indeed common in unsupervised learning tasks, **unsupervised semantic segmentation (USS) poses additional challenges compared to unsupervised image classification**. In USS, neither image-level nor pixel-level labels are available, so only **pairwise pixel relations** can be exploited. As prior studies (Refs[1-3]) demonstrate, architectural designs often play a minor role; the key novelty lies in the **learning strategy for semantic structures under fully unsupervised constraints.**
>
> In this context, our work introduces significant innovations in this context through two major contributions.
>
> (1) In affinity-structure Learning, we formulate pixel-wise semantic relation discovery as a **learnable binary classification problem**, where soft pair labels are distilled from the clustered label space.
>
> Unlike prior USS methods, which typically learn pixel-pair relations implicitly via objectives such as contrastive similarity between pixel or region features (Refs[2]),  consistency or spatial-coherence regularization in feature or neighborhood space (Refs[1]), mutual information maximization over paired predictions (Refs[4]), k-NN or threshold-based pseudo-positive pair mining from pretrained embeddings (Refs[3]). Our proposed DS$^2$DL explicitly models pixel-wise semantic relations as a classification task. Furthermore, our affinity-structure branch incorporates **cross-space distillation** and **reversed directional mining**, which fundamentally differentiates it from prior pairwise or contrastive approaches.
>
> (2) In cluster-structure learning, we adopt a Local-to-Global Code Prediction (LGP) strategy to capture global semantic groupings. Our **global prototypes** summarize the dataset-level semantic structure and serve as global semantic coordinates, similar in spirit to prior USS works that introduce dataset-level concept or prototype structures to stabilize global grouping (e.g., concept clusterbooks in CAUSE (Refs[5]) or EM-fitted class prototypes in PriMaPs-EM (Refs[6])). At the same time, **local prototypes** aggregate region-level features within each image, acting as more stable spatial semantic anchors than raw pixel embeddings.
>
>  Crucially, instead of using global prototypes merely for assignment or consistency regularization (Refs[5,6]) or discovering global groups via multiview hierarchical clustering (Refs[7])), our L2G introduces a prediction-based constraint that forces locally-derived pixel-level cluster codes to predict the global cluster structure. This yields an explicit **bidirectional semantic alignment** between local patch semantics and global dataset-level semantics, going beyond prior uses of global structures.
>
> In summary, our approach provides **novel solutions in pixel-pair relation modeling, local-to-global semantic alignment, cross-space distillation, and reversed directional mining**, all specifically designed for unsupervised semantic segmentation. These innovations extend beyond simply assembling existing components and demonstrate clear methodological originality.
>
> References：
> [1] PiCIE: Unsupervised Semantic Segmentation Using Invariance and Equivariance in Clustering, CVPR 2021.
> [2] Unsupervised Semantic Segmentation by Distilling Feature Correspondences (STEGO), ICLR 2022.
> [3] Leveraging Hidden Positives for Unsupervised Semantic Segmentation, CVPR 2023.
> [4] InfoSeg: Unsupervised Semantic Image Segmentation with Mutual Information Maximization, DAGM GCPR 2021.
> [5] CAUSE: Causal Unsupervised Semantic Segmentation, NeurIPS 2023.
> [6] PriMaPs-EM: Boosting Unsupervised Semantic Segmentation with Principal Mask Proposals, ECCV 2024.
> [7] Unsupervised Hierarchical Semantic Segmentation with Multiview Cosegmentation and Clustering Transformers (HSG), CVPR 2022.

---

> ### Author Response · Authors · 2025-11-26
> **Official Comment by Authors**
>
> **Q2:** Lack of acknowledgment and discussion of classical methods such as binary pairwise losses and OT-based coding, creating the impression that these techniques originated in this work.
>
> **A2:** We thank the reviewer for the valuable comment and the opportunity to clarify the distinctions with prior works. We acknowledge that prior works have proposed related ideas, and we have cited several relevant references in the original manuscript, including SwAV (Refs [8]), OT/Sinkhorn (Refs [9]), and SphereFace2 (Refs [10]) (an earlier version of pairwise similarity learning). In the revised manuscript, we will additionally cite SimPLE (Refs [11]).
> We would like to clarify the differences and our specific contributions:
>
> (1) **Binary classification for pairwise similarity (related to Eq. (1)):** While Eq. (1) uses a binary form similar to SimPLE (Refs [11]), it is **not a direct adoption**. SimPLE is built for generic instance-level pairwise similarity learning—such as image retrieval or re-identification—where ground-truth class labels are available. In contrast, our setting is fundamentally different: we operate at the pixel level and the supervision is fully unsupervised, as required in unsupervised semantic segmentation. Moreover, the binary targets in our method are **soft pair labels distilled from the clustered label space**, rather than hard positives or negatives defined by defined instance identities. Our affinity-structure branch thus introduces a **learnable affinity classifier** with segmentation-specific pair mining, which is tailored for dense, pixel-level predictions.
>
> (2) **Online coding with OT (related to Eq. (5)):** SwAV (Refs [8]) introduced online clustering using OT to learn **image-level** representations. In contrast, our Local-to-Global (L2G) code prediction uses EMA-updated global semantic prototypes to supervise pixel- and patch-level codes, forming a hierarchical semantic bridge across pixel → patch → global levels. This hierarchy arises because global prototypes guide patch representations, which in turn refine pixel-level codes, creating a top–down and bottom–up alignment pathway essential for stable pixel grouping.
>
> SwAV, by comparison, performs prototype assignments only at the image level, without propagating semantic structure to fine-grained spatial representations.
>
> In summary, although prior works provide conceptual precedents for binary similarity objectives and OT-based online clustering, our methods adapt and extend these ideas specifically for **unsupervised semantic segmentation**, introducing **pixel-level affinity modeling**, **local-to-global semantic alignment**, **cross-space distillation, and reversed directional mining**. These innovations go beyond merely applying existing techniques and are critical for achieving effective dense prediction under fully unsupervised settings.
>
> References:
> [8] Unsupervised Learning of Visual Features by Contrasting Cluster Assignments (SwAV), NeurIPS 2020.
> [9] Sinkhorn Distances: Lightspeed Computation of Optimal Transportation Distances (Sinkhorn OT), NeurIPS 2013.
> [10] Sphereface2: Binary classiffcation is all you need for deep face recognition, ICLR 2021.
> [11] Pairwise Similarity Learning is SimPLE, ICCV 2023.

---

> ### Author Response · Authors · 2025-11-26
> **Official Comment by Authors**
>
> **Q3:** “The sampling mining function may require further proof. The authors propose to emphasize more on the negative pair loss and decrease the importance of positive pairs. But the intuition is not explicitly explained in Sec. 3.3 and other part of the manuscript.”
>
> **A3:**  In fully unsupervised semantic segmentation, positive pixel pairs are often noisy, especially near semantic boundaries where pixel features are uncertain. In contrast, negative pairs are generally more reliable because pixels from different semantic regions rarely share high similarity. Treating positive and negative pairs symmetrically in hard mining can therefore lead to unstable training dominated by noisy positive pairs.
>
> To address this, our **Reversed Directional Mining** strategy explicitly leverages the **asymmetry** between positive and negative pairs. By computing the signed distance $ t = S(x_i, x_j) + b$ to the decision boundary, we observe that hard positives and hard negatives lie on opposite sides of the boundary. The loss is designed to **downweight hard positives** and **emphasize hard negatives**, thereby reducing the influence of unreliable positive pairs and avoiding gradient cancellation.
>
> We further provide **empirical evidence** in Table 2. Removing negative pairs (“w/o negative sample pairs (mostly corresponding to cross-image pixel pairs)”) or positive pairs (“w/o positive sample pairs (mostly corresponding to intra-image pixel pairs)”) leads to substantial drops in both Accuracy and mIoU, confirming that the balanced use of positive and negative pairs is critical for stable training.
>
> Additionally, the effect of this strategy is visualized in **Fig. 2** and **Fig. 7**, demonstrating how reversed directional mining improves training stability and affinity learning. In **Fig. 4**, we also provide a sensitivity study on the mining strength r, showing that our method consistently benefits from the proposed mining strategy across datasets and adapts to data complexity.
>
> The gradient behavior of r (Appendix A.3) demonstrates this effect analytically.
> In summary, the **theoretical intuition**, **experimental results**, and **visualizations** together support the proposed sampling mining strategy. These explanations are detailed in Section 3.3, Appendix A.3, and the associated figures, clarifying the rationale behind emphasizing negative pairs over positive ones for fully unsupervised semantic segmentation.
> We hope this addresses the reviewer’s concern and provides a clear understanding of our design.
>
> ---
>
> **Q4:** “Requests an ablation to show the benefit of using soft label similarity instead of hard labels in Eq.(1).”
>
> **A4:**  We provide both theoretical motivation and new ablation experiments to demonstrate the benefit of using **soft label similarity** rather than **hard labels** in Eq. (1).
>
> **Theoretically**, soft labels are more suitable for fully unsupervised semantic segmentation because pseudo labels are highly unreliable at the early stage of training, especially for pixels near semantic boundaries. Hard labels tend to introduce incorrect positive pairs, which propagates noise through the pairwise loss in Eq. (1)-(3). In contrast, the soft label similarity $L_{x_i}\cdot L_{x_j}$ provides a probabilistic affinity that naturally downweights uncertain or ambiguous pixels. This leads to more stable pairwise supervision and is consistent with the design of our affinity structure learning module, which relies on reliable positive and negative signals.
>
> **Experimentally**, we conducted an ablation comparing soft and hard labels. The results clearly show the advantage of using soft label similarity:
> | Method | Accuracy | mIoU  |
> |-----------------------------------|----------|-------|
> | Hard Labels                | 28.99    | 4.95  |
> | Soft Labels (step < 2000), then Hard Labels | 81.67    | 19.33 |
> | Soft Labels  | 84.61    | 20.75 |
>
> Using hard labels from the beginning results in severe performance degradation due to noisy pseudo labels. Even when switching to hard labels only after 2000 steps of soft-label training, the performance is still inferior to using soft labels throughout. This confirms that soft labels provide more stable and effective supervision during affinity structure learning.
>
> In summary, soft labels help prevent noise amplification at early training stages, better reflect pixel-level uncertainty, and integrate naturally with our reversed directional mining strategy. We have incorporated the motivation for adopting soft labels and the corresponding theoretical analysis into **Section 3.3** and **Appendix A.2**, and have added the ablation results and discussions to **Section 4.4** in the revised manuscript. We hope this clarification fully addresses the reviewer’s concern and clearly demonstrates the benefits of using soft label similarity.

---

### Official Review · Reviewer_PZZW · 2025-10-31

**Soundness:** 3
**Presentation:** 3
**Contribution:** 2
**Rating:** 6
**Confidence:** 3

**Summary:**

The paper proposes a new framework for unsupervised semantic segmentation that improves both local consistency and global coherence without using labeled data. The method juses a two-part architecture that models pairwise pixel relationships through a binary similarity classification, and a cluster level loss, which aligns per-pixel predictions with global prototypes. This dual self-distillation transfers semantic knowledge from label space to feature space, ensuring both fine-grained boundary accuracy and consistent global grouping. They use entropy-regularizers and cluster orthogonality constraints on prototypes as well. The work boasts significant performance gains over previous state-of-the-art methods across COCO-Stuff, Cityscapes, and Potsdam datasets.

**Strengths:**

- The paper introduces a variety of interesting new loss functions to help improve unsupervised semseg performance
- The paper compares against a wide variety of baselines and benchmarks on commonly used benchmark tasks
- Good ablations and hyperparameter exploration
- Nice summary of prior work and good descriptions of the added methods

**Weaknesses:**

- Perhaps compare  against or comment some recent open domain semabtic segmentation methods like those that use diffusion of VLMs
- Possibly explore some other backbones to understand dependence on backbone choice
- The paper introduces a "bag of tricks" style of additional loss terms, is there a way to unify some of them or reduce unecessary ones?

**Questions:**

- How sensitive is the loss to different weightings of the loss terms?

---

> ### Author Response · Authors · 2025-11-26
> **Official Comment by Authors**
>
> We sincerely thank Reviewer PZZW for the careful review and insightful questions. We respectfully address each concern below:
>
> ***
>
> **Q1:** “Perhaps compare against or comment some recent open domain semantic segmentation methods like those that use diffusion of VLMs.”
>
> **A1:** Thank you for the helpful suggestion. In the revised manuscript, we have added a dedicated discussion section on recent diffusion- and VLM-based open-domain semantic segmentation approaches (e.g., ODISE (Refs[1]), PnP-OVSS (Refs[2])).
>
> However, these approaches fall outside the scope of our problem setting. Methods based on diffusion models or VLMs rely heavily on strong external semantic priors, such as CLIP text–image alignment, class-name prompts, or pretrained diffusion guidance. In contrast, our method operates in a strictly unsupervised, closed-set scenario without any textual, multimodal, or pretrained semantic supervision. Because the objectives, assumptions, and evaluation protocols differ fundamentally, direct comparison would not be meaningful.
>
> We view these approaches as complementary rather than competing. Our goal is to advance structural representation learning within the unsupervised regime, while open-domain methods focus on leveraging external semantic knowledge. As noted in the manuscript, integrating diffusion- or VLM-based semantic cues is a promising direction for future work.
>
> References:
> [1] ODISE: Open-Vocabulary Object Detection and Segmentation with Diffusion Models, ICCV 2023.
> [2] PnP-OVSS: Emergent Open-Vocabulary Segmentation from Off-the-Shelf VLMs, CVPR 2024.
>
> ---
> **Q2:** “Possibly explore some other backbones to understand dependence on backbone choice.”
>
> **A2:** Thank you for the insightful suggestion. To investigate whether our method depends on a specific backbone architecture, we further evaluated **ResNet-50** as an alternative backbone, initialized with MoCo v2 pre-training, under the same training protocol on the COCO-Stuff dataset. As shown in the table below, our method consistently improves over corresponding baselines, demonstrating that the performance gains are attributed to our proposed dual-structure self-distillation rather than reliance on a particular backbone or pre-training scheme.
>
>
>
> | Methods         | Backbone   | Acc. | mIoU |
> |-----------------|------------|------|------|
> | MoCoV2          | ResNet-50  | 25.2 | 10.4 |
> | + STEGO         | ResNet-50  | 43.1 | 19.6 |
> | + SmooSeg       | ResNet-50  | 52.4 | 18.8 |
> | + DS$^2$DL (Ours)          | ResNet-50  | **53.5** | **20.1** |
>
> The smaller absolute gains on ResNet-50 (MoCo v2) compared to ViT/8 (DINO) can be explained by their distinct inductive biases:
>
> (1) DINO ViT leverages global self-attention, enabling every token to interact with all others. This produces rich long-range affinities and structural cues, which align well with our affinity- and cluster-structure distillation modules;
>
> (2) ResNet-50 (MoCo v2) builds context via local convolutions with gradually expanding receptive fields, resulting in weaker region-level affinity signals. This limits the magnitude of improvement, though improvements remain consistent.
>
> These results collectively show that our method is architecture-agnostic and benefits both convolutional and transformer-based backbones, while naturally achieving larger gains when the backbone provides stronger global contextual features.

---

> > ### Author Response · Authors · 2025-11-26
> > **Official Comment by Authors**
> >
> > **Q3:** “The paper introduces a ‘bag of tricks’ style of additional loss terms, is there a way to unify some of them or reduce unnecessary ones?”
> >
> > **A3:** Thanks for the constructive comment. We clarify that the introduced loss terms are **not independent “bag-of-tricks” components**, but arise naturally from the **structural requirements** of our framework. Each term enforces a specific constraint essential for stable learning:
> >
> > (1) $L_{L2G}$ aligns local pixel features with global semantic prototypes to ensure consistent semantic assignment across images.
> > (2) $L_{orth}$ maintains mutual orthogonality among prototypes to prevent prototype collapse.
> > (3) $L_{data}$ provides dataset-level regularization to stabilize instance-prototype interactions and prevent semantic drift.
> > (4) $L_{p}$ enforces affinity-structure distillation to capture pairwise relations.
> >
> > As shown in **Table 2**, removing any of these components leads to clear drops in both Acc. and IoU, confirming that each term plays a necessary structural role.
> >
> > For clarity, the first three losses can be viewed as **cluster-structure regularization loss**, i.e.,
> >
> > $L_{\text{clus}} = L_{L2G} + L_{\text{orth}} + L_{\text{data}}$.
> >
> > Combined with the affinity-structure learning loss $L_p$ , the overall objective can be summarized as:
> >
> > $L_{\text{total}} = L_{\text{p}} + L_{\text{clus}}$,
> >
> > This grouping reflects **functional coherence** rather than arbitrary aggregation. In other words, these terms are already organized into a unified, principled objective, and none of them is redundant or unnecessary.
> >
> > ---
> >
> > **Q4:**“How sensitive is the loss to different weightings of the loss terms?”
> >
> > **A4:** Thank you for the insightful comment. As explained in our previous response regarding the composition of the total objective, our overall loss consists of the cluster-structure regularization loss and the affinity-structure distillation loss:
> >
> > $L_{\text{total}} = L_{\text{p}} + L_{\text{clus}}$,
> >
> > To evaluate sensitivity to loss weighting, we introduce a scaling factor `λ` on the clustering loss:
> >
> > $L_{\text{total}} = L_{\text{p}} + \lambda L_{\text{clus}}$,
> >
> > We tested different values of `λ` (from 0.2 to 5) on the Cityscapes dataset. The results are shown below:
> >
> > | λ     | Accuracy | mIoU   |
> > |-------|----------|--------|
> > | 0.2   | 82.9649  | 20.5523 |
> > | 0.4   | 83.0437  | 20.5909 |
> > | 0.6   | 83.3610  | 20.6151 |
> > | 0.8   | 84.0609  | 20.7147 |
> > | 1.0   | **84.6127** | **20.7530** |
> > | 1.2   | 84.0609  | 20.7151 |
> > | 1.6   | 84.0611  | 20.7149 |
> > | 2.0   | 84.0610  | 20.7148 |
> > | 5.0   | 84.0643  | 20.7183 |
> >
> > As observed, both mIoU and Accuracy remain highly stable across a wide range of `λ`, with maximum variations of 0.2% and 1.65%, respectively. This demonstrates that our method is **not sensitive to precise loss weighting**, and the overall performance remains robust.
> >
> > We will include this sensitivity analysis in the revised manuscript.

---

### Author Response · Authors · 2025-12-03
**Response to all reviewers**

We sincerely thank all reviewers for their thoughtful and constructive feedback. We are pleased that the reviewers recognized our method as **highly original** (Reviewer Tm7S), **impressive performance** (Reviewer e8JZ and x2DC) and **solid methodology** (Reviewer PZZW).

___

Our work is the first to explicitly and jointly model the affinity (local) and cluster (global) structures for unsupervised semantic segmentation within a unified self-distillation framework, as recognized by Reviewer Tm7S. The main contributions of our work are as follows:

(1) Pixel-level affinity modeling via semantic-space distillation.

We explicitly model **pixel-wise** semantic relations as a **learnable binary classification problem** supervised by **soft pair labels** distilled from the clustered label space. The **Reversed Directional Mining** (RDM) strategy stabilizes the learning process by reducing reliance on noisy positive samples and emphasizing negative samples.

(2) Cluster-Level Structure Distillation (CLSD) for cross-level semantic alignment.

Unlike SwAV, which clusters local features into prototypes, CLSD distills global cluster distributions into local features, enabling local predictions to be guided by global semantic structures and achieving **bidirectional local-to-global semantic alignment**.

---

To address reviewers’ questions and further strengthen the paper, we conducted additional analyses and experiments, summarized below:

- **RDM vs. standard BCE**: Replacing RDM with standard BCE caused significant performance drops, highlighting RDM's importance for stable learning. The analysis on RDM are detailed in Section 3.3 and Appendix A.3.

- **Soft vs. hard label supervision**: Hard labels led to substantial degradation in performance, while soft labels improved learning stability. Using hard labels from the beginning results in severe performance degradation due to noisy pseudo labels. Even when switching to hard labels only after 2000 steps of soft-label training, the performance is still inferior to using soft labels throughout. This confirms that soft labels provide more stable and effective supervision during affinity structure learning.

- **Backbone generality experiments (ResNet-50)**: We demonstrated architecture independence, showing that our framework works consistently across different backbone architectures (CNN and Transformer).

- **Loss-weight sensitivity study**: We examined the impact of weighting between the affinity-structure distillation loss $L_{\text{p}}$ and the cluster-structure regularization loss $L_{\text{clus}} = L_{\text{L2G}} + L_{\text{orth}} + L_{\text{data}}$, by evaluating the weighted total objective: $L_{\text{total}} = L_{\text{p}} + \lambda \ L_{\text{clus}}$. We tested different values of $\lambda $ on the Cityscapes dataset, ranging from 0.2 to 5. The results show that both mIoU and Accuracy remain highly stable, varying by only 0.2% and 1.65%, respectively. These results show that our method is robust to loss weighting, and the unified loss is clearly presented in Section 3.5.



These results collectively reinforce that the observed improvements stem from the core principles of our formulation.

___

We sincerely thank the reviewers again for their valuable feedback. Their insights significantly strengthened the clarity, rigor, and completeness of our work.

---

### Note · Program_Chairs · 2026-01-17
**Submission Desk Rejected by Program Chairs**

The following references in this submission do not refer to real documents and/or have major errors in bibliographic information:

 Hanxiao Chang, Tian Wang, Qifan Meng, and Yan Li. Active bias: Training more accurate neural networks by emphasizing high variance samples. In NeurIPS, 2017.